# Generalized Boundary FDR Control under Arbitrary Dependence: An Approach on Closure Principle

**Yifan Zhang** [*1]   **Wentao Zhang** [*1]   **Changliang Zou** [2]   **Haojie Ren** [1]

## Abstract

False discovery rate (FDR) is a cornerstone of modern multiple testing. However, it often fails to guarantee the reliability of "marginal" discoveries that lie at the boundary of the rejection set, which are often crucial in high-precision applications. While recent works (Soloff et al., 2024; Xiang et al., 2025) introduced the boundary false discovery rate (bFDR) to control the error probability at the marginal discovery, their method relies on restrictive assumptions such as independence or specific prior distributions. In this paper, we first propose $k$-bFDR, a novel generalization that controls the error probability of the $k$ least significant discoveries. We then provide a systematic investigation into the theoretical relationship between $k$-bFDR and existing error metrics. Furthermore, building upon the closure principle, we develop Domino, a unified framework that guarantees $k$-bFDR control under arbitrary dependence, applicable for both p-values and e-values. We prove the theoretical validity of the proposed Domino algorithm and demonstrate through extensive numerical experiments that it consistently achieves rigorous $k$-bFDR control while identifying trustworthy marginal discoveries. Analyses of real data reveal that $k$-bFDR control yields higher-quality rejection sets with greater practical significance.

## 1. Introduction

In the era of high-throughput data analysis, the ability to identify true signals through thousands of hypotheses is fundamental to modern scientific discovery and technological innovation. This refers to multiple hypothesis testing, an important field in statistics that aims to control the extent of false discoveries among rejected hypotheses. The most widely used and classical metric for this purpose is the false discovery rate (FDR) introduced by Benjamini & Hochberg (1995), which provides a global, expectation-based assessment of the rejection set. However, a well-known yet often overlooked limitation of FDR is that it only guarantees the *average quality* of the rejection set.

In high-precision applications such as large-scale genomic studies, one requires reliability for each individual discovery, especially those with the least evidence in the rejection set. For instance, in pan-cancer cohorts, standard FDR-control procedures (e.g., the BH procedure in (Benjamini & Hochberg, 1995)) frequently identify canonical genes such as TP53 mutations with infinitesimal p-values (Lawrence et al., 2014). However, the presence of such exceptionally strong signals leads to a "hitchhiking" phenomenon: some weaker, marginal candidates are also included in the rejection set without sufficient evidence. The mechanism underlying this phenomenon stems from the fact that FDR (see definition in (1)) concerns the proportion of false rejections among the final rejection set. Concretely, the rejection threshold of the BH method is proportional to the size of the final rejection set (i.e., $|\mathcal{R}|\alpha/m$; see notations in Section 2 below). When a large number of strong signals are rejected, the rejection set expands, thereby relaxing the threshold. This relaxation in turn allows weak-evidence individuals, which in fact belong to the null hypothesis, to enter the rejection set. In essence, these marginal candidates "hitchhike" into the discovery set on the back of the expansion driven by strong signals. As a result, although the overall FDR is controlled at the nominal level, the individuals at the boundary of the rejection set are often false discoveries, thus compromising the local reliability of the rejection set. As illustrated in Figure 1, even when the BH procedure successfully controls FDR at level $\alpha = 0.2$, up to 8 out of the last 10 rejections at the boundary may be false discoveries. Since scientific discovery often hinges on validating these novel, borderline candidates rather than reconfirming the well-established strong signals, this lack of boundary control undermines the statistical validity and reproducibility

---

[*]Equal contribution [1]School of Mathematical Sciences, Shanghai Jiao Tong University, Shanghai, China [2]School of Statistics and Data Sciences, LPMC, KLMDASR and LEBPS, Nankai University, Tianjin, China. Correspondence to: Haojie Ren <haojieren@sjtu.edu.cn>.

*Proceedings of the 43$^{rd}$ International Conference on Machine Learning*, Seoul, South Korea. PMLR 306, 2026. Copyright 2026 by the author(s).

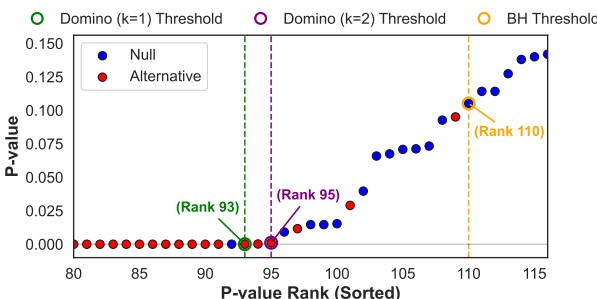

*Figure 1.* Comparison of rejection sets at the boundary across proposed Domino under bFDR and $k$-bFDR ($k = 2$) control and BH procedure under FDR control in DepMap CRISPR-Cas9 screening dataset (Tsherniak et al., 2017). Red and blue points are essential genes and non-essential genes, respectively.

of new findings. Similar risks arise in some high-stakes applications, such as portfolio selection, where a single non-performing stock at the decision boundary can disproportionately compromise overall returns. In these scenarios, the *average quality* of the discovery set guaranteed by FDR is no longer enough, since global error metrics fail to capture the potential risk of boundary uncertainty, especially when the cost for a false rejection is severe. Therefore, in such scenarios, it is necessary to control additional or alternative metrics that focus on the reliability of these boundary discoveries.

Recent works (Soloff et al., 2024; Xiang et al., 2025) introduced a novel error metric, the boundary false discovery rate (bFDR), defined as the probability that the last rejected hypothesis is a false discovery. By shifting from the average error rate (FDR) to the one near the boundary (bFDR), bFDR provides better error control for situations requiring consistent precision across all rejections. Despite its conceptual elegance, existing methods for bFDR control remain limited. Soloff et al. (2024); Xiang et al. (2025) established the Support Line (SL) procedure with provable bFDR control for p-values under the independence assumptions. Gao et al. (2026) extended the SL method to tackle the issue of conformal novelty detection for bFDR control. However, in practice, researchers often need to validate a small batch of marginal discoveries rather than a single one, all while dealing with complex, unknown dependence structures.

**Contributions.** In this paper, we conduct a comprehensive investigation of error control at the boundary of rejection sets. We first introduce $k$-bFDR, a novel generalization of the boundary error rate that quantifies the joint error probability of the $k$ least promising discoveries in the rejection set. Unlike the original bFDR, $k$-bFDR allows researchers to calibrate their risk tolerance for a batch of marginal discoveries. Building upon this metric, we systematically study

the resulting $k$-bFDR framework and its relationships with existing metrics, such as FWER and FDR.

Furthermore, we develop the Domino algorithm, a versatile framework that guarantees $k$-bFDR control under arbitrary dependence structures while remaining compatible with both p-values and e-values. Drawing inspiration from the classical closure principle (Marcus et al., 1976; Bretz et al., 2009), Domino constructs the "closure" tests tailored to the marginal set of the rejection set, which differs from its conventional application for FWER control. Domino is compatible with both p-values and e-values, offering a principled approach to boundary control. Numerical experiments show that the Domino algorithm achieves valid $k$-bFDR control under complex dependence structures. Studies on real-world datasets further indicate that the proposed method yields high-quality rejection sets with practical utility.

## 2. Generalized Metric: $k$-Boundary FDR

In this section, we formally define the $k$-bFDR and consider it within the broader landscape of error metrics in multiple testing. We begin with the problem setup and commonly used error metrics.

Consider $m$ null hypotheses $\mathbb{H}_1, \ldots, \mathbb{H}_m$, and define $\theta_j = 0/1$ if $\mathbb{H}_j$ is true/false for $j \in [m]$. Denote the set of true nulls as $\mathcal{H}_0 = \{j : \theta_j = 0\}$. To assess the evidence against each $\mathbb{H}_j$, we consider an observable test statistic, such as a p-value $p_j$ or e-value $e_j$. To be concrete, for one null hypothesis $\mathbb{H}_j$, its p-value $p_j$ satisfies $\mathbb{P}(p_j \leq u) \leq u$ for all $u \in [0, 1]$ if $\theta_j = 0$, while its e-value $e_j$ (Vovk & Wang, 2021; Wang & Ramdas, 2022) satisfies $\mathbb{E}[e_j] \leq 1$ if $\theta_j = 0$. We reject the null hypothesis for a small p-value or large e-value. A multiple testing procedure aims to construct a rejection set $\mathcal{R} \subseteq [m]$ based on the observed statistics that maintains proper control over a specified error metric.

### 2.1. Error Metrics

A classical error metric is the family-wise error rate (FWER), defined as the probability of making more than one false rejection:

$$\text{FWER}(\mathcal{R}) := \mathbb{P}\left(|\mathcal{H}_0 \cap \mathcal{R}| \geq 1\right).$$

Lehmann & Romano (2005) further generalized the FWER concept to the $k$-FWER to control the probability of making $k$ false rejections for $1 \leq k \leq m$, defined as:

$$k\text{-FWER}(\mathcal{R}) := \mathbb{P}\left(|\mathcal{H}_0 \cap \mathcal{R}| \geq k\right).$$

The classical procedures to control FWER include the Bonferroni correction (Bonferroni, 1936; Dunn, 1961), the Holm procedure and their generalized versions (Lehmann & Romano, 2005), as well as the closed testing framework (Marcus et al., 1976; Guo & Rao, 2010). While these methods

provide rigorous FWER control under arbitrary dependence, they are often overly conservative, especially for large $m$.

The FDR introduced by Benjamini & Hochberg (1995) offers an alternative perspective that focuses on the expected proportion of false discoveries among rejected hypotheses. Formally, the FDR of rejection set $\mathcal{R}$ is

$$\mathrm{FDR}(\mathcal{R}) := \mathbb{E}\left[\frac{|\mathcal{H}_0 \cap \mathcal{R}|}{|\mathcal{R}| \vee 1}\right]. \qquad (1)$$

The emphasis of FDR on error proportion control renders it well-suited for large-scale multiple testing. This characteristic has contributed to various methodological developments in FDR-based approaches, including control procedures under different dependence structures (Benjamini & Yekutieli, 2001), adaptive implementation schemes (Storey, 2002), and methods from a Bayesian perspective (Sun & Cai, 2007). Due to its enhanced power while maintaining interpretability, FDR has become the most widely adopted error metric in large-scale testing and leads to numerous extensions such as the marginal FDR (Storey, 2003) for technical convenience, and the false discovery exceedance (FDX) (Genovese & Wasserman, 2004; 2006) for tail probability guarantees.

Despite their popularity, both FDR and its extensions are global metrics that quantify the *average* error proportion within the rejection set. They do not safeguard the reliability of discoveries at the rejection boundary, where the risk of "hitchhiking" nulls is high. Soloff et al. (2024) has shifted focus to assessing the reliability of marginal rejections and developed a rejection procedure based on independent p-values. Xiang et al. (2025) further formally introduced the boundary FDR (bFDR) concept and established a rigorous formulation within the p-value context. Here, we refine the original bFDR formulation to a unified representation.

**Definition 2.1** (bFDR). Let $I_{\mathcal{R}}^{(1)}$ be the index of the "least significant" discovery in $\mathcal{R}$ (e.g., the one with the largest p-value or smallest e-value), and $I_{\varnothing}^{(1)} = 0 \notin \mathcal{H}_0$ by convention. The bFDR is defined as:

$$\mathrm{bFDR}(\mathcal{R}) := \mathbb{P}\left(I_{\mathcal{R}}^{(1)} \in \mathcal{H}_0\right). \qquad (2)$$

The bFDR shows the probability that the last rejected hypothesis $\mathbb{H}_{I_{\mathcal{R}}^{(1)}}$ is a false discovery. The definition in (2) aligns with that of Xiang et al. (2025), while additionally accommodating the e-value scenario. While bFDR provides a necessary marginal guarantee, existing control methods (such as the SL procedure in (Soloff et al., 2024)) typically require independence of p-values, which limits their applicability in complex machine learning tasks.

## 2.2. $k$-Boundary FDR

While bFDR focuses on the single most marginal hypothesis in the rejection set, practical applications often require

evaluating the joint reliability of a small batch of borderline rejections. We propose a more general metric, $k$-bFDR, which is defined as the probability of falsely rejecting the hypotheses corresponding to the "$k$ least significant" discoveries in $\mathcal{R}$, where $1 \le k \le m$.

**Definition 2.2** ($k$-bFDR). Let $I_{\mathcal{R}}^{(k)}$ be the index of the $k$-th "least significant" discovery in $\mathcal{R}$ (e.g., the one with $k$-th largest p-value or $k$-th smallest e-value). The $k$-bFDR is defined as

$$k\text{-bFDR}(\mathcal{R}) := \mathbb{P}\left(\left\{I_{\mathcal{R}}^{(1)}, \ldots, I_{\mathcal{R}}^{(k)}\right\} \subseteq \mathcal{H}_0\right).$$

By convention, $k$-bFDR is 0 for any rejection set with fewer than $k$ elements (i.e., $|\mathcal{R}| < k$) since such a set cannot commit $k$ false discoveries simultaneously. This aligns with the fact that the bFDR of an empty set is 0.

When $k = 1$, the $k$-bFDR reverts to the conventional bFDR. By extending the focus from solely on the most marginally rejected hypothesis to the $k$ least significant rejections, the $k$-bFDR captures higher-order uncertainty among borderline discoveries. This generalization provides researchers with enhanced flexibility to tailor the stringency of error evaluation according to specific application requirements.

We now present several simple yet important properties of the $k$-bFDR and its relationship with classical error metrics.

**Proposition 2.3.** *(a) Under the global null, i.e., when all null hypotheses are true, $k$-bFDR is equivalent to $k$-FWER. In particular, bFDR is equivalent to the FDR. (b) In general settings, $k$-bFDR is no larger than $k$-FWER.*

The proof of Proposition 2.3 and any other necessary proofs will be detailed in Appendix A. Proposition 2.3 demonstrates that under the global null, bFDR-type metrics exhibit consistent interpretation with traditional FWER-type control. More broadly, in general settings, the $k$-bFDR serves as a more relaxed error criterion than $k$-FWER, allowing for enhanced power. Consequently, $k$-bFDR provides an alternative metric for multiple hypothesis testing, effectively balancing error control with increased power while maintaining clear probabilistic interpretations.

The relationship between bFDR and FDR exhibits fundamental complexity (Xiang et al., 2025). Gao et al. (2026) found that the FDR theoretically is no larger than the bFDR under a monotonicity assumption, where non-null p-values tend to be smaller. However, the validity of the monotonicity assumption in practical applications remains uncertain. When monotonicity is violated, the rejection at the boundary characterized by bFDR may not necessarily correspond to the theoretically least compelling scenario. Rather than comparing these metrics in isolation, we further investigate the procedures for controlling them through the lens of the closure principle in Section 4.

# 3. Domino: $k$-bFDR Control Procedure

In this section, we introduce the Domino framework for $k$-bFDR control. We first present the core algorithm and its theoretical guarantees, followed by a discussion of its practical implementation and comparisons to other methods.

## 3.1. Control of $k$-bFDR

The closure principle is a fundamental pillar in multiple testing, providing a systematic way to construct procedures with rigorous control of error metrics under arbitrary dependence (Marcus et al., 1976; Bretz et al., 2009; Xu et al., 2025). Inspired by this principle, we propose a novel procedure named Domino to control $k$-bFDR. Unlike traditional closed testing, Domino focuses on the boundary of discoveries and is compatible with both p-values and e-values to enhance its practical applicability.

The building block of our framework is the $k$-local test. For any index subset $S \subseteq [m]$ with $|S| \geq k$, the intersection null hypothesis is $\mathbb{H}_S := \cap_{j \in S} \mathbb{H}_j$. A valid $k$-local test is required to safeguard against the inclusion of $k$ or more false discoveries within the subset $S$.

**Definition 3.1** (Valid $k$-local test). For a subset $S \subseteq [m]$ with $|S| \geq k$, let $\phi_S^k \in \{0, 1\}$ be a test statistic for $\mathbb{H}_S$ with $\phi_S^k = 1$ indicating significance if and only if at least $k$ of the individual hypotheses are found significant. We say $\phi_S^k$ is a **valid $k$-local test** at level $\alpha$ if it satisfies $\mathbb{P}(\phi_S^k = 1) \leq \alpha$.

*Remark* 3.2. The closed testing was originally developed for p-value-based testing (Guo & Rao, 2010; Goeman & Solari, 2011) and has been extended to accommodate e-value-based testing methods (Hartog & Lei, 2025; Xu et al., 2025). Note that Definition 3.1 is agnostic to the underlying type of evidence. The only essential condition is that the probability of falsely declaring significance does not exceed $\alpha$ under the intersection null.

With the definition of the $k$-local test, we now introduce the Domino condition. To streamline our presentation, we first focus on the scenario using p-values and then discuss the implementation with e-values. Consider p-value $p_j$ associated evidence for hypothesis $\mathbb{H}_j$ for $j \in [m]$, and let $\pi : \{1, \ldots, m\} \to \{1, \ldots, m\}$ be a permutation that sorts the p-values such that $p_{\pi(1)} \leq p_{\pi(2)} \leq \cdots \leq p_{\pi(m)}$.

For a candidate rejection size $r$, we define the candidate rejection set as $\mathcal{R}_r = \{\pi(1), \ldots, \pi(r)\}$. The corresponding marginal index set $\mathcal{M}_{r,k}$, standing for the $k$ least significant discoveries in $\mathcal{R}_r$, is,

$$\mathcal{M}_{r,k} = \{\pi(r-k+1), \pi(r-k+2), \ldots, \pi(r)\}. \quad (3)$$

The Domino procedure identifies the largest $r$ such that every intersection hypothesis containing $\mathcal{M}_{r,k}$ is rejected by a valid $k$-local test.

---

**Algorithm 1** Domino based on p-values (Domino-P)

1: **Input:** null hypotheses $\mathbb{H}_1, \ldots, \mathbb{H}_m$ with observed p-values $p_1, \ldots, p_m$, target $k$-bFDR level $\alpha$, valid $k$-local test $\phi_S^k$ for $S \subseteq [m]$.
2: Sort the p-values in ascending order $p_{\pi(1)} \leq p_{\pi(2)} \leq \cdots \leq p_{\pi(m)}$;
3: Initialize the rejection set $\mathcal{R} = \{j : p_j \leq p_{\pi(k-1)}\}$ and the index $r = m$;
4: **while** $|\mathcal{R}| < k$ and $r \geq k$ **do**
5:     Let $\mathcal{M}_{r,k} = \{\pi(r-k+1), \pi(r-k+2), \ldots, \pi(r)\}$;
6:     **if** $\mathcal{C}(r)$ in (4) holds for $\mathcal{M}_{r,k}$ **then**
7:         Update $\mathcal{R} = \{j : p_j \leq p_{\pi(r)}\}$;
8:         **break**
9:     **end if**
10:    Update $r \leftarrow r - 1$;
11: **end while**
12: **Output:** rejection set $\mathcal{R}$.

---

**Definition 3.3** (Domino condition). A candidate size $r$ satisfies the $k$-boundary closure condition if, for any subset of indices $S \subseteq [m]$ that contains the marginal set $\mathcal{M}_{r,k}$, the valid $k$-local test $\phi_S^k$ rejects:

$$\mathcal{C}(r) : \phi_S^k = 1, \text{ for all } S \supseteq \mathcal{M}_{r,k}. \quad (4)$$

We focus on the rank-consecutive marginal set $\mathcal{M}_{r,k}$ in (3) since its failure to satisfy (4) implies that no other non-consecutive size-$k$ set containing $\pi(r-k+1)$ as the minimal-rank element can fulfill the condition. If all $r \geq k$ fail to fulfill (4) in Definition 3.3, the final rejection set reduces to the trivial set with $k-1$ most significant hypotheses: $\mathcal{R} = \{j \in [m] : p_j \leq p_{\pi(k-1)}\}$ with $p_{\pi(0)} := 0$.

The whole Domino algorithm based on p-values is summarized in Algorithm 1. Following a similar sketch, the Domino algorithm based on e-values is detailed in Appendix B. The Domino algorithm focuses on determining the boundary of the rejection set to achieve $k$-bFDR control, as it depends only on these boundary elements. The core lies in identifying the marginal index set that meets the specified condition in (4), and then rejecting all hypotheses that the corresponding statistics indicate more significance than these boundary ones, akin to toppling a chain of dominoes where pushing the initial dominoes triggers the fall of all subsequent ones.

Applying the closure principle, the probability of falsely rejecting the marginal set satisfying (4) can be proved to be bounded, thereby achieving $k$-bFDR control for the whole rejection set. The following theorem establishes strict $k$-bFDR control guarantees for the proposed Domino method.

**Theorem 3.4.** *The rejection set $\mathcal{R}$ produced by the Domino algorithm satisfies $k$-$\mathrm{bFDR}(\mathcal{R}) \leq \alpha$.*

Theorem 3.4 shows that Domino achieves rigorous $k$-bFDR control without requiring any structural assumptions on the dependence among p-values or e-values. In the specific case of $k = 1$, Domino provides a robust guarantee for bFDR control under arbitrary dependence. This distinguishes our approach from the SL procedure in (Soloff et al., 2024), which primarily ensures bFDR control under the independence assumption for p-values. By leveraging the closure principle, our framework not only extends the applicability to e-values but also broadens the applicability of the bFDR control in more general and complex settings.

### 3.2. Choices of Valid $k$-Local Tests

The Domino framework is flexible, allowing for various forms of the $k$-local test $\phi_S^k$ that guarantee the probability of falsely rejecting $\mathbb{H}_S$ at $\alpha$. It is necessary to select a valid $k$-local test $\phi_S^k$ depending on the type of evidence available and the structural assumptions.

Under arbitrary dependence, a straightforward option is the generalized Bonferroni method (Lehmann & Romano, 2005), which employs an adjusted critical threshold of $k\alpha/|S|$ in p-value-based testing for $k$-FWER control, i.e.,

$$\phi_S^{k,\text{Bonf}} = 1 \iff \frac{|S|}{k} p_{(k:S)} \leq \alpha, \qquad (5)$$

where $p_{(k:S)}$ denotes the $k$-th smallest p-value in subset $S$. The generalized Holm procedure (Lehmann & Romano, 2005) for $k$-FWER control can also serve as a valid $k$-local test, with decision rules identical to (5). Additionally, if the p-values are assumed independent or certain dependence structures, more effective tests can be employed. For example, under independence or positive regression dependence on a subset (PRDS; Benjamini & Yekutieli, 2001), the Simes test (Simes, 1986) provides a valid local test for $k = 1$:

$$\phi_S^{1,\text{Simes}} = 1 \iff \min_{1 \leq j \leq |S|} \frac{|S|}{j} p_{(j:S)} \leq \alpha.$$

Its generalized variant (Sarkar, 2008) can be employed as a valid $k$-local test, resulting in a more powerful procedure.

When conducting hypothesis testing using e-values, we develop a valid $k$-local test by the e-closure principle. Specifically, for a subset $S \subseteq [m]$ with $|S| \geq k$, let

$$\phi_S^{k,\text{e-closure}} = 1 \iff \exists\, W_S \subseteq S \text{ with } |W_S| \geq k, \text{ such that}$$
$$e_T \geq \frac{\mathbb{1}\{|T \cap W_S| \geq k\}}{\alpha} \text{ for all } T \subseteq S, \qquad (6)$$

where $e_T$ denotes the e-value for $\mathbb{H}_T$.

**Proposition 3.5.** *The test $\phi_S^{k,\text{e-closure}}$ in (6) is a valid $k$-local test at level $\alpha$.*

For the special case $k = 1$, a valid local test $\phi_S^1$ can simplify to aggregate the p-values or e-values within the subset $S$ into

a single statistic to test $\mathbb{H}_S$. Under arbitrary dependence, Vovk & Wang (2020) explored the combination of multiple p-values using generalized means and demonstrated that the scaled harmonic mean of p-values for $|S| \geq 2$ is valid:

$$\phi_S^{1,\text{Harmo}} = 1 \iff e \ln |S| \frac{|S|}{\sum_{j \in S} \frac{1}{p_j}} \leq \alpha, \qquad (7)$$

where $e$ is the Euler's number and the scaling factor $e \ln |S|$ ensures validity. For the case when $|S| = 1$, the test can be performed directly using the single p-value itself. If independence is assumed, the Cauchy combination method in (Liu & Xie, 2020) provides a more powerful approach for p-value aggregation to capture sparse signals more effectively. In the context of e-values, some straightforward combination strategies are allowed for local testing. Under arbitrary dependence, the simple averaging approach could constitute a valid 1-local test (Vovk & Wang, 2021). That is,

$$\phi_S^{1,\text{eAve}} \iff \frac{1}{|S|} \sum_{j \in S} e_j \geq 1/\alpha.$$

Under independence, the multiplicative aggregation of e-values can lead to significant power gains (Xu et al., 2025).

### 3.3. Computational Implementations of Domino

At first glance, verifying the Domino condition $\mathcal{C}(r)$ in (4) presents a heavy computational challenge, as it requires checking $2^{m-k}$ possible subsets $S \supseteq \mathcal{M}_{r,k}$. However, for most common choices of $k$-local tests, this procedure can be significantly simplified in practice. By leveraging the ordering of test statistics and the structural properties of the test $\phi_S^k$, we need only verify the condition in (4) for a reduced subset collection $\{S\}$.

For example, consider the generalized Bonferroni procedure as the $k$-local test $\phi_S^{k,\text{Bonf}}$, which only checks the $k$-th smallest p-value in the subset $S$. For a marginal set $\mathcal{M}_{r,k}$, let its $k$-th smallest p-value be $p_{\pi(r)}$. Any subset $S$ formed by augmenting $\mathcal{M}_{k,r}$ with "weaker" hypotheses (those with indices $\{\pi(r+1), \ldots, \pi(m)\}$) will keep $p_{\pi(r)}$ as $k$-th order statistic, leaving the decision criteria unchanged, thus obviating the need for additional verification. Consequently, it suffices to verify the condition only for subsets $S$ constructed by sequentially adding elements from $\{\pi(1), \ldots, \pi(r-k)\}$ in reverse order to $\mathcal{M}_{r,k}$. This reduces the exponential search over $S$ to a linear search, and the whole algorithm is detailed in Appendix C.1.

We develop computationally efficient implementations of Domino in Algorithm 1 for cases where the $k$-local test employs either the generalized Bonferroni procedure $\phi_S^{k,\text{Bonf}}$ or the p-value combination by scaled harmonic mean $\phi_S^{1,\text{Harmo}}$. As demonstrated in the Appendix C, these optimizations reduce the computational complexity from exponential order $O(2^m)$ to a tractable polynomial order $O(m^2)$.

## 4. Relationship to Existing Paradigms

In this section, we first discuss the relationship between the proposed Domino and closed testing for $k$-FWER control, followed by a formal connection to the e-closure principle (Xu et al., 2025) and finally, a comparison between bFDR and FDR control through the lens of rejection set properties.

**Domino v.s. closed testing.** In closed testing for $k$-FWER control, a hypothesis $\mathbb{H}_j$ is rejected if (Guo & Rao, 2010)

$$\phi_S^k = 1, \text{ for all } S \ni j \text{ with } |S| \geq k \text{ satisfying } p_j \geq p_{(k:S)}.$$

This implies that enforcing $k$-FWER control requires every individual element in the rejection set to adhere to the closure principle across all relevant subsets. In contrast, Domino for achieving $k$-bFDR control only requires the $k$ marginal elements of the rejection set $\mathcal{R}$ to satisfy the closure condition (4). Consequently, Domino is less conservative than $k$-FWER-controlling procedures, leading to a larger rejection set while maintaining boundary reliability. This distinction in rejection patterns reflects the fundamental shift in the error metric: while $k$-FWER ensures the entire rejection set against any $k$ false discoveries, $k$-bFDR focuses its statistical budget on the least significant part of the discovery list.

**Connection to e-closure principle.** The e-closure principle (Xu et al., 2025) has recently emerged as a unified framework for controlling both expectation-based (e.g., FDR) and tail-probability-based error rates using e-values. Since the $k$-bFDR can be expressed in expectation form as $k\text{-bFDR}(\mathcal{R}) = \mathbb{E}\left[\mathbb{1}\left\{\left\{I_{\mathcal{R}}^{(1)}, \ldots, I_{\mathcal{R}}^{(k)}\right\} \subseteq \mathcal{H}_0\right\}\right]$, the e-closure principle framework can also achieve $k$-bFDR control. We find that when taking $\phi_S^{k,\text{e-closure}}$ in (6) as the $k$-local test in Domino, it is equivalent between Domino and the e-closure principle in the sense that two principles yield identical rejection sets.

**Proposition 4.1.** *Consider the error rate function $f_S^{k\text{-bFDR}}(\mathcal{R}) = \mathbb{1}\left\{\left\{I_{\mathcal{R}}^{(1)}, \ldots, I_{\mathcal{R}}^{(k)}\right\} \subseteq S\right\}$ in the e-closure principle, the e-closure principle is identical to Domino when taking $\phi_S^{k,\text{e-closure}}$ in (6) as the $k$-local test.*

As a result, the e-closure principle for $k$-bFDR control can be considered as a special case of the Domino framework. Furthermore, one can anticipate that Domino's modularity allows for the integration of diverse $k$-local tests that may strictly outperform the standard e-closure principle.

**Discussion of bFDR and FDR.** We further elucidate the distinction between the two criteria, bFDR and FDR, by carefully analyzing the rejection process through the e-closure principle to gain deeper insights into their distinct quality control paradigms. Under this framework, the rejection set $\mathcal{R}^{\text{bFDR}}$ for bFDR control satisfies

$$e_S \geq 1/\alpha \text{ for all } S \ni I_{\mathcal{R}^{\text{bFDR}}}^{(1)},$$

where $e_S$ is the e-value for $\mathbb{H}_S$. Meanwhile, the rejection set $\mathcal{R}^{\text{FDR}}$ for FDR control must satisfy

$$e_S \geq \frac{|\mathcal{R}^{\text{FDR}} \cap S|}{|\mathcal{R}^{\text{FDR}}|} \frac{1}{\alpha} \text{ for all } S \cap \mathcal{R}^{\text{FDR}} \neq \varnothing.$$

These differing requirements highlight the distinct emphases of bFDR and FDR. Concretely, the bFDR control targets only the least significant element in the rejection set by enforcing a fixed threshold of $1/\alpha$, emphasizing *marginal quality*. In contrast, the FDR control reflects the *average quality* by employing an adaptive threshold for every element in the rejection set weighted by the proportion of rejected hypotheses within $S$, which is generally smaller than $1/\alpha$. Moreover, this discount threshold in FDR control allows highly significant signals to "mask" marginal errors, resulting in those hitchhiking marginal discoveries. Thus, bFDR-type control provides a more robust solution for applications where boundary uncertainty carries high risk.

## 5. Numerical Experiments

In this section, we evaluate the performance of the Domino framework through synthetic simulations and real data analysis. We focus on three primary metrics: (i) $k$-bFDR, which assesses marginal reliability; (ii) true discovery rate (TDR), defined as the expected proportion of true positives among rejections, reflecting the discovery quality; (iii) Power, defined as the expected proportion of true positives that are correctly rejected, which reports the ability to detect true signals. All results are evaluated over 100 replications. The implementation code for the numerical experiments in this paper is available at https://github.com/WenntaoZhang/Domino.

Unless specified in the subsequent analysis, we adopt the following configurations: **Domino-P** ($k = 1$) employs the Simes local test when the null p-values are independent or PRDS, and switches to the Harmonic mean p-value combination for arbitrary or unknown dependence; **Domino-P** ($k > 1$) uses the generalized Bonferroni $k$-local test; **Domino-E** adopts the $k$-local test developed from the e-closure principle and uses the arithmetic average to combine e-values.

### 5.1. Synthetic Data

We generate data from a multivariate normal distribution $\boldsymbol{X} = (X_1, \cdots, X_m) \sim \mathcal{N}(\boldsymbol{\mu}, \boldsymbol{\Sigma})$. The null hypotheses are $\mathbb{H}_j : \mu_j \leq 0$ for $j \in [m]$. The true state $\theta_j \in \{0, 1\}$ follows

*Table 1.* The bFDR, TDR (%), and Power (%) of different Domino methods at $\alpha = 0.05$, $\mu_c = 3$, and $\pi_1 = 0.2$.

| Local test | $\rho = 0$ | | | $\rho = 0.25$ | | |
|---|---|---|---|---|---|---|
| | $k$-bFDR | TDR | Power | $k$-bFDR | TDR | Power |
| $\phi^{1,\text{Simes}}$ | 0.00 | 98.9 | 42.7 | 0.01 | 99.4 | 41.9 |
| $\phi^{1,\text{Harmo}}$ | 0.00 | 99.8 | 27.0 | 0.00 | 99.9 | 26.5 |
| $\phi^{1,\text{eAVG}}$ | 0.00 | 99.5 | 25.0 | 0.00 | 100.0 | 24.4 |
| $\phi^{2,\text{Bonf}}$ | 0.00 | 98.9 | 48.4 | 0.01 | 99.4 | 47.6 |
| $\phi^{2,\text{e-closure}}$ | 0.00 | 99.8 | 33.9 | 0.00 | 100.0 | 32.9 |
| $\phi^{3,\text{Bonf}}$ | 0.00 | 98.9 | 51.9 | 0.00 | 99.2 | 51.5 |
| $\phi^{3,\text{e-closure}}$ | 0.00 | 99.5 | 40.6 | 0.00 | 99.7 | 39.3 |

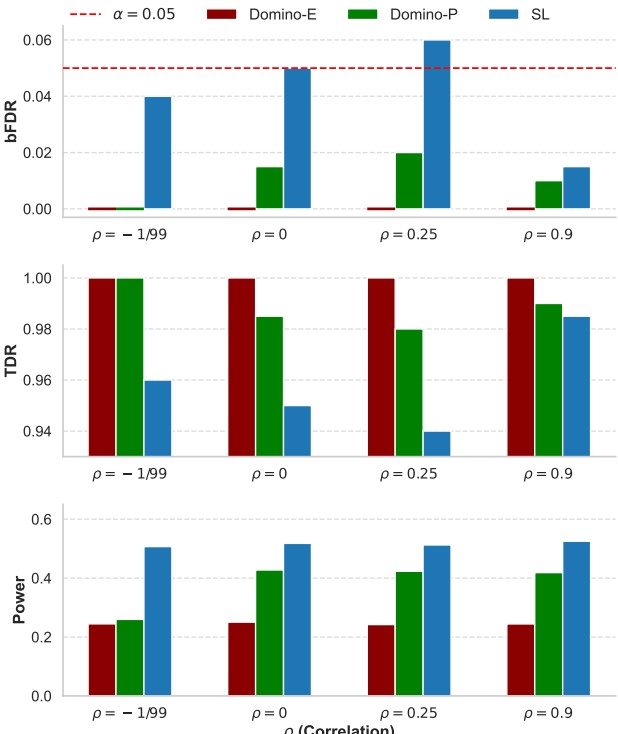

*Figure 2.* The bFDR, TDR, and Power of Domino-P, Domino-E, and SL under different correlation structures.

a latent Bernoulli distribution $\theta_j \sim \text{Bern}(\pi_1)$, where $\pi_1$ is the sparsity hyperparameter. For null hypotheses, the true mean is set to be $\mu_j = 0$; for alternatives, the signal strength $\mu_j$ is i.i.d from a normal distribution $\mathcal{N}(\mu_c, \sigma^2)$ truncated to $(0, +\infty)$. The covariance matrix $\Sigma$ is configured with $\Sigma_{jj} = \sigma^2$ and $\Sigma_{ij} = \sigma^2\rho$ for $i \neq j$, where $\rho \in [-\frac{1}{m-1}, 1)$ is the correlation coefficient. The p-values are calculated based on a one-sided Z-test $p_j = \Phi(-X_j/\sigma)$, while the e-values are set as the likelihood ratio $e_j = \exp\{\frac{\mu_c X_j - 0.5\mu_c^2}{\sigma^2}\}$. Unless otherwise specified, we set $m = 100$, $\sigma = 1$, the hyperparameter $\pi_1 = 0.2$ and the nominal level $\alpha = 0.05$.

We first evaluate the performance of Domino across different local tests for $k$-bFDR control with $k \in \{1, 2, 3\}$ in Table 1 under independence ($\rho = 0$) and weak positive correlation ($\rho = 0.25$). As illustrated, all Domino variants successfully control the corresponding $k$-bFDR while maintaining a high TDR. Under the bFDR constraint, using the Simes-based local test $\phi^{1,\text{Simes}}$ yields the superior power by exploiting the independence structure of the p-values. The Domino using local test $\phi^{1,\text{Harmo}}$ is inherently less sensitive to i.i.d settings; however, it offers the distinct advantage of maintaining validity across arbitrary dependence structure, as demonstrated in Table 5. Domino-E methods yield a bit more conservative, thus safer, rejection set than the p-value-based counterparts. In terms of relaxed boundary error metric ($k \in \{2, 3\}$), both Domino-P and Domino-E yield rejection sets with higher power, illustrating the tradeoff between precise, high-quality discoveries and detection ability.

We further compare Domino against SL (Soloff et al., 2024) from negative correlation to strong positive correlation. As illustrated in Figure 2, the SL method suffers from a bFDR inflation under weak positive correlation ($\rho = 0.25$). Domino, in contrast, not only ensures robust bFDR control under all tested dependence structures but also outperforms SL in TDR. This advantage stems from the closure-based search mechanism of Domino, which more effectively navigates the discovery boundary under complex dependencies. Meanwhile, the Domino methods, especially Domino-P,

achieve power comparable to that of the SL method.

Additional simulation results varying complex correlation structures, sparsity $\pi_1$, nominal level $\alpha$, and signal strength $\mu_c$ are provided in Appendix E.

### 5.2. Real Data: CRISPR Gene Discovery

To evaluate the performance of our proposed Domino, we utilize a public genomic CRISPR-CAS9 screening dataset from the Cancer Dependency Map (DepMap) project (Tsherniak et al., 2017). The dataset consists of gene effect scores for each cell line, quantifying the impact of knocking out a specific gene on the survival of the cell line. A highly negative score implies that the gene is essential for cell survival, while a score near zero is prone to be non-essential. Our goal is to identify a reliable gene set where all the selected genes are survival-essential.

We formulate a one-sided hypothesis test where the null hypothesis assumes the gene effect score is 0, while the alternative assumes the score is negative. To calculate the p-values and error rate, we leverage two standardized reference gene datasets from the DepMap portal: one is a set of genes universally recognized as vital, and the other is considered negligible for cell survival. We extract a representative cell line, known as ACH-000011 (NCI-H2087),

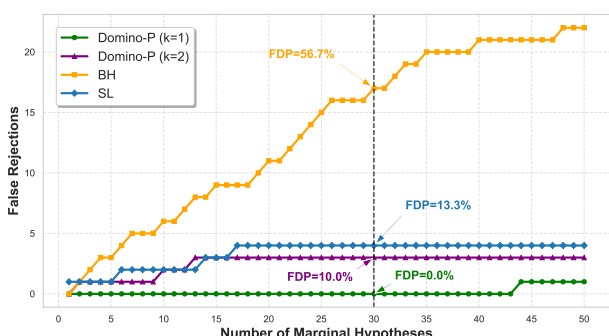

*Figure 3.* Comparison of rejection sets within the top 50 marginal rejections at the boundary across different methods at nominal level $\alpha = 0.1$. The FDP corresponding to the top 30 marginal rejections for each method is also displayed.

and map the genes to these two reference datasets. 657 genes and 688 genes are assigned binary labels as vital and non-essential, respectively. The p-values are calculated by learning the distribution of the scores in the non-essential reference genes.

We compare the performance of the BH procedure against three bFDR-control methods, including the SL, Domino-P ($k = 1$), and Domino-P ($k = 2$). The nominal level is set at $\alpha = 0.1$. To quantify boundary reliability, we investigate the number of false discoveries among the 50 least significant rejections for each method in Figure 3. It is demonstrated that the BH procedure suffers from severe boundary error inflation, where nearly half of its marginal discoveries are false positives. This confirms our motivation: in the presence of extremely strong signals, global FDR control allows low-quality candidates to "hitchhike" into the rejection set. In contrast, Domino-P ($k = 1$) selects the most reliable and high-quality candidates, maintaining high precision on the marginal hypotheses with negligible boundary errors. By increasing the marginal size to $k = 2$, Domino-P ($k = 2$) yields a slightly larger rejection set with an increased tolerable boundary error, which remains below the target level. This implies that our framework offers the flexibility for researchers who are more risk-tolerant. The SL method produces a slightly lower-quality discovery set compared to the Domino framework. These results highlight Domino's ability to protect the discovery boundary, ensuring that borderline candidates remain statistically trustworthy.

We further evaluated the bFDR, TDR, and power in a re-sampling experiment over 100 runs, where in each run one-quarter of the data were randomly allocated to estimate the distribution for computing p-values and the remaining three-quarters were reserved for testing. As shown in Table 2, the Domino framework consistently achieves the lowest bFDR

while simultaneously maintaining the highest TDR. This indicates that the rejection sets identified by Domino possess superior reliability and precision compared to other competing methods. Though the power of Domino-P is slightly lower than that of SL, SL suffers a bFDR inflation under dependence. Table 3 presents the performance of the $k$-bFDR Domino-P procedure across $k \in \{1, 2, 3\}$. As expected, a larger $k$ implies a more liberal constraint on the boundary error metric, allowing the researchers to select a specific configuration aligned with their risk tolerance and discovery objectives. The relevant results for Domino-E are presented in Appendix F.1.

## 5.3. Real Data: S&P 500 Stock Selection

We apply our method to the high-stakes stock selection task, utilizing the constituents of the S&P 500 index downloaded from Yahoo Finance. As a proxy for the U.S. large-cap equity market, this dataset presents a challenging environment where signal-to-noise ratios are often low. We consider the stock data from 2024 to 2025 and exclude assets with missing data or those delisted during the period. Our goal is to construct a high-quality set of stocks with high returns and low risk.

We employ the Capital Asset Pricing Model (CAPM) to estimate the return of each stock. For the stock $j$, its daily return is modeled as $R_{j,t} = \alpha_j + \beta_j R_{M,t} + \epsilon_{j,t}$, where $R_{M,t}$ is the return of benchmark (S&P 500 here). The intercept $\alpha_j$ represents the return potential of stock $j$ over the S&P 500. The null hypothesis for each stock is $\mathbb{H}_j : \alpha_j \leq 0$. The one-sided p-values are constructed by t-statistics of the estimated $\hat{\alpha}_j$ using 2024 data. Portfolio selection is performed based on these p-values, and an equally-weighted "Buy-and-Hold" strategy is simulated throughout the out-of-sample period in 2025.

To avoid selection bias, we conducted a randomized re-sampling study, selecting 20 stocks per trial over 100 independent repetitions. The average number of selections and strategy returns are summarized in Table 4. The results demonstrate that Domino successfully picks out the most valuable and credible set of stocks, outperforming both SL and BH in terms of portfolio quality. This outcome serves as a microcosm of real-world high-stakes scenarios. When the penalty for a poor investment choice is substantial, statistical guarantees regarding the decision boundary are often of far greater significance.

## 6. Summary

This work establishes a comprehensive framework for error control at the rejection boundary through the introduction of $k$-bFDR and the Domino algorithm. By shifting the focus from global average quality to localized marginal reliability,

*Table 2.* The bFDR, TDR (%), and Power (%) of three methods based on p-values at different nominal levels over 100 repetitions.

| Method | $\alpha = 0.05$ | | | $\alpha = 0.1$ | | | $\alpha = 0.2$ | | |
|---|---|---|---|---|---|---|---|---|---|
| | bFDR | TDR | Power | bFDR | TDR | Power | bFDR | TDR | Power |
| Domino | 0.04 | 99.37 | 86.49 | 0.06 | 99.32 | 87.20 | 0.06 | 99.27 | 87.89 |
| SL | 0.19 | 98.84 | 90.42 | 0.27 | 98.67 | 90.93 | 0.32 | 98.48 | 91.45 |
| BH | 0.46 | 96.48 | 93.71 | 0.59 | 95.18 | 94.78 | 0.66 | 92.35 | 96.19 |

*Table 3.* The $k$-bFDR, TDR (%), and Power (%) of Domino-P with different $k$ at different nominal levels over 100 repetitions.

| Domino-P | $\alpha = 0.05$ | | | $\alpha = 0.1$ | | | $\alpha = 0.2$ | | |
|---|---|---|---|---|---|---|---|---|---|
| | $k$-bFDR | TDR | Power | $k$-bFDR | TDR | Power | $k$-bFDR | TDR | Power |
| $k = 1$ | 0.04 | 99.37 | 86.49 | 0.06 | 99.32 | 87.20 | 0.06 | 99.27 | 87.89 |
| $k = 2$ | 0.03 | 98.98 | 89.76 | 0.06 | 98.79 | 90.37 | 0.04 | 98.64 | 90.97 |
| $k = 3$ | 0.00 | 98.88 | 90.13 | 0.02 | 98.69 | 90.73 | 0.01 | 98.52 | 91.29 |

*Table 4.* The average number of selected stocks and returns (%) of three methods at different nominal levels over 100 repetitions.

| Method | $\alpha = 0.1$ | | $\alpha = 0.2$ | |
|---|---|---|---|---|
| | Number | Return | Number | Return |
| Domino ($k = 1$) | 9.35 | 12.42 | 9.39 | 12.41 |
| Domino ($k = 2$) | 9.79 | 12.36 | 9.85 | 12.38 |
| Domino ($k = 3$) | 9.80 | 12.38 | 9.91 | 12.35 |
| SL | 9.62 | 11.94 | 9.69 | 11.97 |
| BH | 9.91 | 12.35 | 9.92 | 12.32 |

we address the critical "hitchhiking" phenomenon that often compromises the reproducibility of borderline discoveries in high-throughput studies. The proposed Domino framework, rooted in a novel adaptation of the closure testing principle, bridges the gap between theoretical rigor and practical utility. Domino can guarantee error control under arbitrary dependence and is compatible with both p-values and e-values. Our empirical results in genomics and finance demonstrate that by protecting the marginal set, Domino effectively filters out boundary noise where traditional FDR procedures fail. As a scalable and modular meta-framework, Domino offers a robust foundation for trustworthy and reproducible discovery in high-stakes applications.

The general theoretical nature of Domino allows it to manage various dependencies and be applicable across diverse practical settings, which entails a potential trade-off in testing power. In particular, when additional information suggests the presence of specific dependence patterns, such as independence, weak dependence, or exchangeable structures, the Domino method may not fully exploit these conditions and structures. As a result, it might have reduced effectiveness in detecting true alternatives compared to methods that are specifically tailored to such structures. This motivates future research on designing more efficient strategies that adapt to known dependence structures while maintaining $k$-bFDR control.

## Acknowledgements

We sincerely thank the anonymous reviewers for their insightful comments and constructive suggestions, which have significantly enhanced the quality of this manuscript. Haojie Ren was supported by the National Key R&D Program of China (Grant No. 2024YFA1012200), and the National Natural Science Foundation of China (Grant Nos. 12471262, 12522115), and Shanghai Jiao Tong University 2030 Initiative. Changliang Zou was supported by the National Key R&D Program of China (Grant No. 2022YFA1003800) and the National Natural Science Foundation of China (Grant No. 12231011).

## Impact Statement

This paper presents work whose goal is to advance the field of machine learning. There are many potential societal consequences of our work, none of which we feel must be specifically highlighted here.

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

# A. Proofs

In this section, we include all the necessary proofs of the results throughout the paper.

## A.1. Proof of Proposition 2.3

*Proof.* (a) When all null hypotheses are true, all rejections must necessarily be false discoveries, yielding $\mathbb{1}\left\{\left\{I_{\mathcal{R}}^{(1)}, \ldots, I_{\mathcal{R}}^{(k)}\right\} \subseteq \mathcal{H}_0\right\} = \mathbb{1}\left\{|\mathcal{H}_0 \cap \mathcal{R}| \geq k\right\}$ and hence $k\text{-bFDR}(\mathcal{R}) = k\text{-FWER}(\mathcal{R})$.

Additionally, building upon the discussion regarding the relationship between FDR and FWER in (Benjamini & Hochberg, 1995), we can readily derive the equivalence between bFDR and FDR under the global null.

(b) In general settings, it holds that $\mathbb{1}\left\{\left\{I_{\mathcal{R}}^{(1)}, \ldots, I_{\mathcal{R}}^{(k)}\right\} \subseteq \mathcal{H}_0\right\} \leq \mathbb{1}\left\{|\mathcal{H}_0 \cap \mathcal{R}| \geq k\right\}$ by definition. Therefore, we have $k\text{-bFDR}(\mathcal{R}) \leq k\text{-FWER}(\mathcal{R})$ after taking expectations on both sides. $\square$

## A.2. Proof of Theorem 3.4

*Proof.* Without loss of generality, assume the size of the rejection set $\mathcal{R}_r$ produced by the Domino algorithm is $r \geq k$ and $\mathcal{R}_r$ satisfies the Domino condition $\mathcal{C}(r)$ in Definition 3.3. Otherwise, only the trivial set with $\mathcal{R}_{k-1}$ most significant hypotheses would be rejected, thereby trivially satisfying $k\text{-bFDR}$.

Assume $|\mathcal{H}_0| \geq k$, or the conclusion holds trivially. For rejection set $\mathcal{R}_r$, it holds that

$$\mathbb{1}\{\mathcal{M}_{k,r} \subseteq \mathcal{H}_0\} \leq \mathbb{1}\{\phi_{\mathcal{H}_0}^k = 1\}$$

by the Domino condition $\mathcal{C}(r)$ in Definition 3.3. Taking expectations on both sides,

$$k\text{-bFDR}(\mathcal{R}_r) = \mathbb{P}\left(\mathcal{M}_{k,r} \subseteq \mathcal{H}_0\right) \leq \mathbb{P}\left(\phi_{\mathcal{H}_0}^k = 1\right),$$

which is bounded by $\alpha$, since $\phi_S^k$ is a valid $k$-local test at level $\alpha$ (Definition 3.1). $\square$

## A.3. Proof of Proposition 3.5

*Proof.* Before demonstrating how the e-closure principle yields valid $k$-local tests, we first establish its capability to control $k$-FWER. Define
$$f_S^{k\text{-FWER}}(\mathcal{R}) = \mathbb{1}\left\{|S \cap \mathcal{R}| \geq k\right\} \text{ for a subset } S \subseteq [m].$$

Taking $f_S^{k\text{-FWER}}(\mathcal{R})$ as the error rate function for $k$-FWER control, if a set $\mathcal{R}$ satisfies

$$\text{for } \mathcal{R}: \ e_S \geq \frac{\mathbb{1}\left\{|S \cap \mathcal{R}| \geq k\right\}}{\alpha} \text{ for all } S \subseteq [m], \tag{8}$$

then $k\text{-FWER}(\mathcal{R}) \leq \alpha$ holds by the e-closure principle (Xu et al., 2025).

We now prove the validity of $\phi_S^{k,\text{e-closure}}$ in (6) as a $k$-local test. In fact, it suffices to show that the condition in (6) for deriving $\phi_S^{k,\text{e-closure}} = 1$ is equivalent to (8). Considering a given subset $S \subseteq [m]$ with $|S| \geq k$, $\phi_S^k = 1$ for testing $\mathbb{H}_S$ if and only if at least $k$ of the individual hypotheses are found significant by definition. By the e-closure principle, this is equivalent to the existence of a subset $W_S \subseteq S$ with $|W_S| \geq k$ satisfying condition (8), i.e.,

$$\exists \, W_S \subseteq S \text{ with } |W_S| \geq k, \text{ such that } e_T \geq \frac{\mathbb{1}\{|T \cap W_S| \geq k\}}{\alpha} \text{ for all } T \subseteq S,$$

which completes the proof. $\square$

## A.4. Proof of Proposition 4.1

*Proof.* Recall that the $k$-bFDR of the rejection set $\mathcal{R}$ can be expressed in expectation form:

$$k\text{-bFDR}(\mathcal{R}) = \mathbb{E}\left[\mathbb{1}\left\{\left\{I_{\mathcal{R}}^{(1)}, \ldots, I_{\mathcal{R}}^{(k)}\right\} \subseteq \mathcal{H}_0\right\}\right] = \mathbb{E}\left[f_{\mathcal{H}_0}^{k\text{-bFDR}}(\mathcal{R})\right],$$

where

$$f_S^{k\text{-bFDR}}(\mathcal{R}) = \mathbb{1}\left\{\left\{I_{\mathcal{R}}^{(1)}, \ldots, I_{\mathcal{R}}^{(k)}\right\} \subseteq S\right\} \text{ for a subset } S \subseteq [m].$$

Taking $f_S^{k\text{-bFDR}}(\mathcal{R})$ as the error rate function in the e-closure principle for $k$-bFDR control, the e-closure principle identifies the largest $r$ such that the marginal set $\mathcal{M}_{r,k}$ of rejection set $\mathcal{R}_r$ satisfies

$$\text{for all } S \supseteq \mathcal{M}_{k,r} : \ e_S \geq \frac{1}{\alpha}. \tag{9}$$

Taking $\phi_S^{k,\text{e-closure}}$ in (6) as the $k$-local test, the Domino identifies the largest $r$ such that the marginal set $\mathcal{M}_{r,k}$ of rejection set $\mathcal{R}_r$ satisfies

$$\text{for all } S \supseteq \mathcal{M}_{k,r} : \ \exists\, W_S \subseteq S \text{ with } |W_S| \geq k, \ \text{such that } e_T \geq \frac{\mathbb{1}\{|T \cap W_S| \geq k\}}{\alpha} \text{ for all } T \subseteq S. \tag{10}$$

To establish the property, it suffices to demonstrate the equivalence between conditions (9) and (10).

First, we prove that condition (9) leads to condition (10). For any given subset $S \supseteq \mathcal{M}_{k,r}$, it holds that $|\mathcal{M}_{k,r}| \geq k$ and for all $T \subseteq S$:

- If $|T \cap \mathcal{M}_{k,r}| < k$, then $e_T \geq \frac{\mathbb{1}\{|T \cap \mathcal{M}_{k,r}| \geq k\}}{\alpha} = 0$ naturally;

- If $|T \cap \mathcal{M}_{k,r}| \geq k$, then we have $T \supseteq \mathcal{M}_{k,r}$ and $e_T \geq \frac{\mathbb{1}\{|T \cap \mathcal{M}_{k,r}| \geq k\}}{\alpha} = \frac{1}{\alpha}$ by (9).

Combining these two cases, we conclude that (10) holds. We now proceed to prove the converse implication. For any given subset $S \supseteq \mathcal{M}_{k,r}$, since (10) holds, there exists $W_S \subseteq S$ with $|W_S| \geq k$, implying $|S \cap W_S| \geq k$. From (10), we have

$$e_S \geq \frac{\mathbb{1}\{|S \cap W_S| \geq k\}}{\alpha} = \frac{1}{\alpha}.$$

Thus, we finish the whole proof. $\square$

## B. Domino Algorithm Based on E-values

In this section, we discuss the scenario of implementing the Domino algorithm based on e-values.

Consider e-value $e_j$ associated evidence for hypothesis $\mathbb{H}_j$ for $j \in [m]$, and let $\pi : \{1, \ldots, m\} \to \{1, \ldots, m\}$ be a permutation that sorts the e-values such that $e_{\pi(1)} \geq e_{\pi(2)} \geq \cdots \geq e_{\pi(m)}$. We then proceed analogously to the p-value-based Domino algorithm to identify the largest $r$ such that every intersection hypothesis containing the marginal set $\mathcal{M}_{r,k}$ defined in (3) is rejected by a valid $k$-local test, i.e., identifying the largest $r$ satisfying the Domino condition in Definition 3.3. If all $r \geq k$ fail to fulfill condition (4), we output the rejection set as $\mathcal{R} = \{j \in [m] : e_j \geq e_{\pi(k-1)}\}$, with $e_{\pi(0)} = +\infty$ by convention. The whole Domino algorithm based on e-values is summarized in Algorithm 2.

The relative advantages of Domino-E over Domino-P stem from the inherent properties of e-values and p-values, as the overall framework of the Domino-E and Domino-P methods is generally aligned. Compared to p-values, e-values are grounded in the concept of expectation and offer a natural advantage in that they are more readily combinable. This makes them potentially easier to integrate into valid $k$-local tests and to yield $k$-local tests with greater power. For instance, we construct a valid $k$-local test based on the e-closure principle as shown in (6). Moreover, Hartog & Lei (2025) demonstrated that the weighted e-Bonferroni tests are provably more powerful than their weighted p-Bonferroni counterparts under the same weights and $(1/e)$-form p-values, indicating that the e-value aggregation is more efficient. Domino-E is also preferred when e-values are easier to construct than p-values, particularly in complex, high-dimensional, or distribution-free settings. For example, for composite null hypotheses, e-values can be formed using mixture likelihood ratios, avoiding the conservative calibration typically required for p-values (Zhang et al., 2024). Furthermore, our simulation results in Section 5.1 and Appendix E indicate that Domino-E is particularly favorable when the goal is to obtain a small but precise rejection set, i.e., a set with a modest number of rejections, nearly all of which are correct. Intuitively, this is because Domino-E tends to reject only hypotheses with sufficiently large e-values. As a result, while Domino-E yields fewer total rejections compared to Domino-P, its TDR approaches one.

---

**Algorithm 2** Domino based on e-values (Domino-E)

---

1: **Input:** null hypotheses $\mathbb{H}_1, \ldots, \mathbb{H}_m$ with observed e-values $e_1, \ldots, e_m$, target $k$-bFDR level $\alpha$, valid $k$-local test $\phi_S^k$ for $S \subseteq [m]$.
2: Sort the e-values in descending order $e_{\pi(1)} \geq e_{\pi(2)} \geq \cdots \geq e_{\pi(m)}$;
3: Initialize the rejection set $\mathcal{R} = \left\{ j : e_j \geq e_{\pi(k-1)} \right\}$ and the index $r = m$;
4: **while** $|\mathcal{R}| < k$ **and** $r \geq k$ **do**
5:     Let $\mathcal{M}_{r,k} = \{\pi(r-k+1), \pi(r-k+2), \ldots, \pi(r)\}$;
6:     **if** $\mathcal{C}(r)$ in (4) holds for $\mathcal{M}_{r,k}$ **then**
7:         Update $\mathcal{R} = \left\{ j : e_j \geq e_{\pi(r)} \right\}$;
8:         **break**
9:     **end if**
10:    Update $r \leftarrow r - 1$;
11: **end while**
12: **Output:** rejection set $\mathcal{R}$.

---

## C. Fast Implementation of Domino

In this section, we develop computationally efficient implementations of Domino based on p-values in Algorithm 1 for cases where the $k$-local test employs either the generalized Bonferroni procedure $\phi_S^{k,\text{Bonf}}$ or the p-value combination by scaled harmonic mean $\phi_S^{1,\text{Harmo}}$.

### C.1. Generalized Bonferroni Procedure

As established in Section 3.3, when employing the generalized Bonferroni procedure as the $k$-local test $\phi_S^{k,\text{Bonf}}$ in (5), for a marginal set $\mathcal{M}_{r,k}$, it suffices to verify the condition only for subsets $S$ constructed by sequentially adding elements from $\{\pi(1), \ldots, \pi(r-k)\}$ in reverse order to $\mathcal{M}_{r,k}$, i.e., first adding $\{\pi(r-k)\}$, then $\{\pi(r-k-1), \pi(r-k)\}$, and finally the full set $\{\pi(1), \ldots, \pi(r-k)\}$. Moreover, as implied by (5), we can initiate the search procedure for $r$ at the largest order statistic that does not exceed $\alpha$, thereby further reducing computational requirements. This whole algorithm, summarized in Algorithm 3, reduces the computational complexity from exponential $O(2^m)$ to polynomial $O(m^2)$.

---

**Algorithm 3** Domino-P with $\phi_S^{k,\text{Bonf}}$ as the $k$-local test

---

1: **Input:** null hypotheses $\mathbb{H}_1, \ldots, \mathbb{H}_m$ with observed p-values $p_1, \ldots, p_m$, target $k$-bFDR level $\alpha$.
2: Sort the p-values in ascending order $p_{\pi(1)} \leq p_{\pi(2)} \leq \cdots \leq p_{\pi(m)}$;
3: Initialize the rejection set $\mathcal{R} = \left\{ j : p_j \leq p_{\pi(k-1)} \right\}$ and the index $r = \arg\max\{v \in [m] : p_{\pi(v)} \leq \alpha\}$;
4: **while** $|\mathcal{R}| < k$ **and** $r \geq k$ **do**
5:     Let $\mathcal{M}_{r,k} = \{\pi(r-k+1), \pi(r-k+2), \ldots, \pi(r)\}$;
6:     Initialize the indicator $\Delta = 1$ and the index $\ell = r$;
7:     **while** $\Delta = 1$ **and** $\ell \geq 1$ **do**
8:         Update $\Delta \leftarrow \Delta \cdot \mathbb{1}\left\{ \frac{k+r-\ell}{k} p_{\pi(\ell)} \leq \alpha \right\}$;
9:         $\ell \leftarrow \ell - 1$;
10:    **end while**
11:    **if** $\Delta = 1$ **then**
12:        Update $\mathcal{R} = \left\{ j : p_j \leq p_{\pi(r)} \right\}$;
13:        **break**
14:    **end if**
15:    Update $r \leftarrow r - 1$;
16: **end while**
17: **Output:** rejection set $\mathcal{R}$.

---

## C.2. P-value Combination by Scaled Harmonic Mean

For the 1-local test $\phi_S^{1,\text{Harmo}}$ in (7) employing p-value combination via scaled harmonic mean, the condition is to check whether the harmonic mean of p-values in the subset scaled by the correction factor $e \ln |S|$ is no larger than $\alpha$. To ensure $\phi_S^{1,\text{Harmo}} = 1$ holds for all $S$ for a marginal set $\mathcal{M}_{r,1} = \{\pi(r)\}$, it reduces to verifying

$$\max_{S \ni \mathcal{M}_{r,1}} e \ln |S| \frac{|S|}{\sum_{j \in S} \frac{1}{p_j}} \leq \alpha.$$

This permits us to focus verification exclusively on critical sets $S$ most likely to violate the condition, i.e., those $S$ where the combined p-values tend to be larger. Importantly, the harmonic mean increases when larger p-values are added to $S$, while the correction factor $e \ln |S|$ also increases strictly with $|S|$. These monotonic properties imply that verification can be optimized by sequentially expanding the marginal set $\mathcal{M}_{r,1}$ through reverse-ordered incorporation of elements from the tail $\{\pi(r+1), \ldots, \pi(m)\}$, requiring only $m - r + 1$ specific conditions in total to be checked. This incremental expansion approach offers an additional computational advantage. The harmonic mean of the augmented set can be computed recursively using the existing harmonic mean of $S$ and the newly added p-value, eliminating the need for complete recomputation over all elements. Specifically, letting $\text{Har}(S) = \frac{|S|}{\sum_{j \in S} \frac{1}{p_j}}$ denote the harmonic mean of p-values in $S$, the updated harmonic mean after adding $p_{\pi(\ell)}$ becomes:

$$\text{Har}(S \cup \{\pi(\ell)\}) = \frac{|S| + 1}{\frac{|S|}{\text{Har}(S)} + \frac{1}{p_{\pi(\ell)}}}.$$

This recursive formulation reduces the computational complexity from linear $O(|S|)$ to constant $O(1)$ per update. Furthermore, analogous to the implementation using the generalized Bonferroni procedure as the $k$-local test in Appendix C.1, we can optimize the search algorithm by initiating the procedure at $r' = \arg\max\{v \in [m] : p_{\pi(v)} \leq \alpha\}$ and $p_{\pi(r')}$ is the largest order statistic satisfying $p_{\pi(r')} \leq \alpha$. The resulting procedure achieves computational efficiency and reduces the total computational complexity from exponential $O(2^m)$ to polynomial $O(m^2)$, which is summarized in Algorithm 4.

---

**Algorithm 4** Domino-P with $\phi_S^{1,\text{Harmo}}$ as the $k$-local test when $k = 1$

1: **Input:** null hypotheses $\mathbb{H}_1, \ldots, \mathbb{H}_m$ with observed p-values $p_1, \ldots, p_m$, target $k$-bFDR level $\alpha$.
2: Sort the p-values in ascending order $p_{\pi(1)} \leq p_{\pi(2)} \leq \cdots \leq p_{\pi(m)}$;
3: Initialize the rejection set $\mathcal{R} = \varnothing$ and the index $r = \arg\max\{v \in [m] : p_{\pi(v)} \leq \alpha\}$;
4: **while** $|\mathcal{R}| < 1$ **and** $r \geq 1$ **do**
5:     Let $\mathcal{M}_{r,1} = \{\pi(r)\}$;
6:     Initialize the indicator $\Delta = 1$, the index $\ell = m$, the subset $S = \mathcal{M}_{r,1}$, and the harmonic mean $\text{Har}(S) = p_{\pi(r)}$;
7:     **while** $\Delta = 1$ **and** $\ell > r$ **do**
8:         Update $S \leftarrow S \cup \{\pi(\ell)\}$;
9:         Update $\text{Har}(S) \leftarrow \frac{|S|}{\frac{|S|-1}{\text{Har}(S)} + \frac{1}{p_{\pi(\ell)}}}$;
10:        Update $\Delta \leftarrow \Delta \cdot \mathbb{1}\{e \ln |S| \text{Har}(S) \leq \alpha\}$;
11:        $\ell \leftarrow \ell - 1$;
12:    **end while**
13:    **if** $\Delta = 1$ **then**
14:        Update $\mathcal{R} = \{j : p_j \leq p_{\pi(r)}\}$;
15:        **break**
16:    **end if**
17:    Update $r \leftarrow r - 1$;
18: **end while**
19: **Output:** rejection set $\mathcal{R}$.

---

# D. Practical Guidance on the Choice of $k$

For practitioners adopting the Domino framework with $k$-bFDR control, the principles guiding the choice of $k$ are fundamentally practice-oriented. Different values of $k$ correspond to distinct error metrics. Selecting an appropriate $k$

is not a matter of determining which criterion is inherently superior. Instead, it depends on the specific risk preferences and decision-making context. From a decision-theoretic perspective, these criteria reflect different risk priorities, and an inappropriate choice may yield results lacking practical significance.

Specifically, the choice of $k$ should be guided by the intended use of the selected set and the tolerance for boundary risk. More generally, $k$ can be interpreted as the size of the "critical frontier" within the rejection set for which reliability is essential, and should therefore align with resource constraints or downstream validation capacity. For instance, if only $k$ candidates can be validated or deployed in practice, setting $k$ accordingly ensures that the entire actionable subset enjoys a rigorous boundary guarantee. This interpretation positions $k$-bFDR as a natural middle ground between global (FDR) and worst-case (FWER) control, offering a principled and application-aligned way to trade off power and reliability.

If one has a downstream task for which the choice of $k$ may lead to varying performance, then a data-driven approach to selecting $k$ would be desirable. However, this problem is highly challenging, as it introduces issues of "double-dipping". Developing a method that enables the selection of $k$ with valid theoretical guarantees would require a carefully redesigned framework, especially given that different downstream tasks may prioritize different performance metrics. As such, this remains an important direction for future research.

## E. Additional Simulation Results

We provide additional simulation results in this section to comprehensively demonstrate the performance characteristics of our proposed Domino method.

### E.1. Domino under Other Correlations

We extend our evaluation to include scenarios with negative correlation ($\rho = -1/99$) and strong positive correlation ($\rho = 0.9$) in Table 5. The configuration of other hyperparameters is the same as Table 1. We also conduct simulations under a stronger signal strength $\mu_c = 6$ in Table 6. From Tables 5 and 6, while $\phi^{1,\text{Simes}}$ is of great discovery capability under i.i.d. and PRDS structure, it may fail to maintain validity under other dependence structures, i.e., negative correlation. Meanwhile, $\phi^{1,\text{Harmo}}$ empirically proves valid bFDR control across arbitrary dependence regimes.

*Table 5.* The $k$-bFDR, TDR(%), and Power (%) of different Domino methods at $\alpha = 0.05$, $\mu_c = 3$, and $\pi_1 = 0.2$ with $\rho \in \{-1/99, 0.9\}$.

| | $\rho = -1/99$ | | | $\rho = 0.9$ | | |
|---|---|---|---|---|---|---|
| Local test | $k$-bFDR | TDR | Power | $k$-bFDR | TDR | Power |
| $\phi^{1,\text{Simes}}$ | 0.00 | 99.8 | 40.9 | 0.02 | 99.1 | 42.9 |
| $\phi^{1,\text{Harmo}}$ | 0.00 | 100.0 | 25.2 | 0.01 | 99.2 | 27.3 |
| $\phi^{1,\text{eAVG}}$ | 0.00 | 100.0 | 23.9 | 0.00 | 100.0 | 24.9 |
| $\phi^{2,\text{Bonf}}$ | 0.00 | 99.7 | 46.0 | 0.01 | 99.8 | 49.9 |
| $\phi^{2,\text{e-closure}}$ | 0.00 | 99.8 | 32.0 | 0.00 | 99.9 | 34.3 |
| $\phi^{3,\text{Bonf}}$ | 0.00 | 98.9 | 49.8 | 0.02 | 99.2 | 54.2 |
| $\phi^{3,\text{e-closure}}$ | 0.00 | 99.9 | 38.6 | 0.01 | 99.7 | 41.1 |

*Table 6.* The $k$-bFDR, TDR(%), and Power (%) of different Domino methods at $\alpha = 0.05$, $\mu_c = 6$, and $\pi_1 = 0.2$.

| | $\rho = -1/99$ | | | $\rho = 0$ | | | $\rho = 0.25$ | | | $\rho = 0.9$ | | |
|---|---|---|---|---|---|---|---|---|---|---|---|---|
| Local test | $k$-bFDR | TDR | Power | $k$-bFDR | TDR | Power | $k$-bFDR | TDR | Power | $k$-bFDR | TDR | Power |
| $\phi^{1,\text{Simes}}$ | 0.06 | 99.67 | 97.45 | 0.02 | 99.76 | 97.42 | 0.05 | 99.77 | 97.56 | 0.00 | 100.00 | 97.60 |
| $\phi^{1,\text{Harmo}}$ | 0.00 | 100.00 | 93.29 | 0.01 | 99.87 | 93.44 | 0.01 | 99.96 | 93.48 | 0.00 | 100.00 | 93.97 |
| $\phi^{1,\text{eAVG}}$ | 0.00 | 100.00 | 88.85 | 0.00 | 99.94 | 89.30 | 0.00 | 100.00 | 90.04 | 0.00 | 100.00 | 90.46 |
| $\phi^{2,\text{Bonf}}$ | 0.00 | 99.48 | 98.03 | 0.00 | 99.60 | 98.33 | 0.04 | 99.52 | 98.36 | 0.00 | 100.00 | 98.25 |
| $\phi^{2,\text{e-closure}}$ | 0.00 | 99.20 | 93.84 | 0.00 | 99.39 | 94.51 | 0.00 | 98.50 | 94.97 | 0.00 | 97.96 | 94.00 |
| $\phi^{3,\text{Bonf}}$ | 0.00 | 99.38 | 98.35 | 0.00 | 99.20 | 98.64 | 0.03 | 99.36 | 98.82 | 0.00 | 99.95 | 98.62 |
| $\phi^{3,\text{e-closure}}$ | 0.00 | 97.05 | 96.90 | 0.00 | 96.82 | 97.36 | 0.00 | 96.19 | 97.75 | 0.00 | 94.91 | 95.86 |

### E.2. Varying Signal Proportion $\pi_1$

We investigate the performance of the methods with $\pi_1$ varying in $\{0.1, 0.2, 0.3, 0.4, 0.5\}$. The nominal level is $\alpha = 0.1$. The signal strength is $\mu_c = 3$. The number of data is $m = 100$, and the correlation coefficient is $\rho = 0.25$. The outcome is displayed in Figure 4. The Domino methods always keep a valid $k$-bFDR control, while the SL method fails to control $k$-bFDR when the proportion $\pi_1$ is small. Moreover, both Domino-P and Domino-E yield rejection sets with higher TDR than SL across all tested alternative proportions. In terms of power, the Domino-P and SL methods exhibit comparable power performance, while Domino-E suffers a power loss due to the conservative nature of e-values.

### E.3. Varying Nominal Level $\alpha$

We investigate the performance of the methods with $\alpha$ varying in $\{0.05, 0.1, 0.15, 0.2, 0.25, 0.3\}$. The alternative proportion is $\pi_1 = 0.2$. The signal strength is $\mu_c = 3$. The number of data is $m = 100$, and the correlation coefficient is $\rho = 0.25$. Figure 5 illustrates that as $\alpha$ increases, the $k$-bFDR and power rise while the TDR declines. Consistent with Figure 4, the Domino methods outperform SL in the sense of maintaining valid $k$-bFDR control while achieving higher TDP and comparable power.

### E.4. Varying Signal Strength $\mu_c$

We investigate the performance of the methods with signal strength $\mu_c$ varying in $\{2, 3, 4, 5, 6, 7\}$ in Figure 6. The alternative proportion $\pi_1 = 0.2$ and the nominal level $\alpha$ is fixed at 0.1. The number of data is $m = 100$, and the correlation coefficient is $\rho = 0.25$. For the case of $k = 1$, while both Domino and SL achieve bFDR control, Domino yields a higher TDR and achieves a relatively high power. When the boundary constraint is relaxed ($k \geq 2$), both Domino-P and Domino-E keep valid $k$-bFDR control. Domino-P remains stable and demonstrates high TDR performance. Domino-E shows a decreasing trend in TDR as $\mu_c$ rises, since the relaxation of the boundary error metric and stronger signals incorporate more rejection counts. In both $k = 1$ and $k \geq 2$ scenarios, the power of Domino-E is slightly lower than that of Domino-P due to the conservativeness of e-values.

## F. Additional Real Data Results

We provide additional real data results in this section to comprehensively demonstrate the performance characteristics of our proposed Domino method.

### F.1. E-value Results on CRISPR Gene Discovery Study in Section 5.2

We compute the likelihood ratio to serve as e-values for testing data after estimating the distribution, while the remaining configuration follows that of the p-value setting in Section 5.2. As summarized in Table 7, the e-value-based results demonstrate that Domino-E gives a high-quality rejection set. It maintains valid control over $k$-bFDR, while giving the highest TDR among all methods evaluated in Section 5.2 with little sacrifice of power. Additionally, the metrics of the Domino-E scale with $k$ in a manner consistent with the trajectory of Domino-P.

*Table 7.* The $k$-bFDR, TDR (%), and Power (%) of Domino-E with different $k$ at different nominal levels over 100 repetitions.

| Domino-E | $\alpha = 0.05$ | | | $\alpha = 0.1$ | | | $\alpha = 0.2$ | | |
|---|---|---|---|---|---|---|---|---|---|
| | bFDR | TDR | Power | bFDR | TDR | Power | bFDR | TDR | Power |
| $k = 1$ | 0.02 | 99.44 | 84.86 | 0.02 | 99.44 | 85.49 | 0.07 | 99.40 | 86.12 |
| $k = 2$ | 0.00 | 99.44 | 85.58 | 0.00 | 99.40 | 86.24 | 0.00 | 99.35 | 86.95 |
| $k = 3$ | 0.00 | 99.42 | 86.08 | 0.00 | 99.38 | 86.74 | 0.00 | 99.33 | 87.43 |

### F.2. Real Data: Pathway Enrich Analysis

We utilize the GSE15852 gene expression dataset (Ni et al., 2010), a classical and widely used benchmark including 43 matched pairs of human breast tumor and normal breast tissues, to further demonstrate the practical gains of the proposed Domino methods in pathways analysis. We adopt the combined score metric of some widely-recognized pathways to evaluate the quality of the discovery set of genes under different methods (Kuleshov et al., 2016). The combined score is

formally defined as $z \log(p)$, where $p$ is the unadjusted Fisher's exact p-value, which is computed on the hypergeometric probability of observing at least $k$ overlapping genes between the rejection set and a known biological pathway, and $z$ is a Z-score. The combined score produces a composite enrichment measure of the quality of rejection set: a discovery set with high combined scores of pathways successfully captures valid, biologically meaningful signals (true positives) while simultaneously exerting strict control over noise (false positives). The combined score is computed using the Enrichr tool (Kuleshov et al., 2016).

We report the combined scores on eight well-studied breast cancer pathways in Table 8 for all competing methods evaluated in Section 5.2. Though the classical BH method rejects three times more than our proposed Domino, it is outperformed by all the bFDR-based methods in pathway enrichment analysis in the sense that it includes too much noise (null). Compared with SL, our methods obtain higher-quality discovery sets that exclude noise precisely while including real biological signals.

*Table 8.* Comparison of combined scores for eight pathways across different multiple testing procedures with $\alpha = 0.05$. Numbers in parentheses indicate the number of rejections.

| Pathway | Domino-P | | | SL (1025) | BH (1769) |
|---|---|---|---|---|---|
| | $k = 1$ (390) | $k = 2$ (431) | $k = 3$ (454) | | |
| PPAR signaling pathway | 301.6 | 259.0 | 290.6 | 173.8 | 111.0 |
| AMPK signaling pathway | 125.2 | 130.6 | 117.9 | 49.5 | 21.9 |
| Fatty acid degradation | 96.8 | 125.9 | 114.8 | 103.5 | 79.0 |
| Regulation of lipolysis in adipocytes | 94.5 | 80.8 | 73.3 | 45.5 | 42.5 |
| Insulin signaling pathway | 60.7 | 50.7 | 45.3 | 24.3 | 9.8 |
| Glycerolipid metabolism | 51.8 | 44.0 | 60.6 | 36.2 | 19.1 |
| ECM-receptor interaction | 26.1 | 48.7 | 43.8 | 45.4 | 60.8 |
| Fc gamma R-mediated phagocytosis | 48.0 | 40.3 | 36.2 | 8.5 | 8.9 |

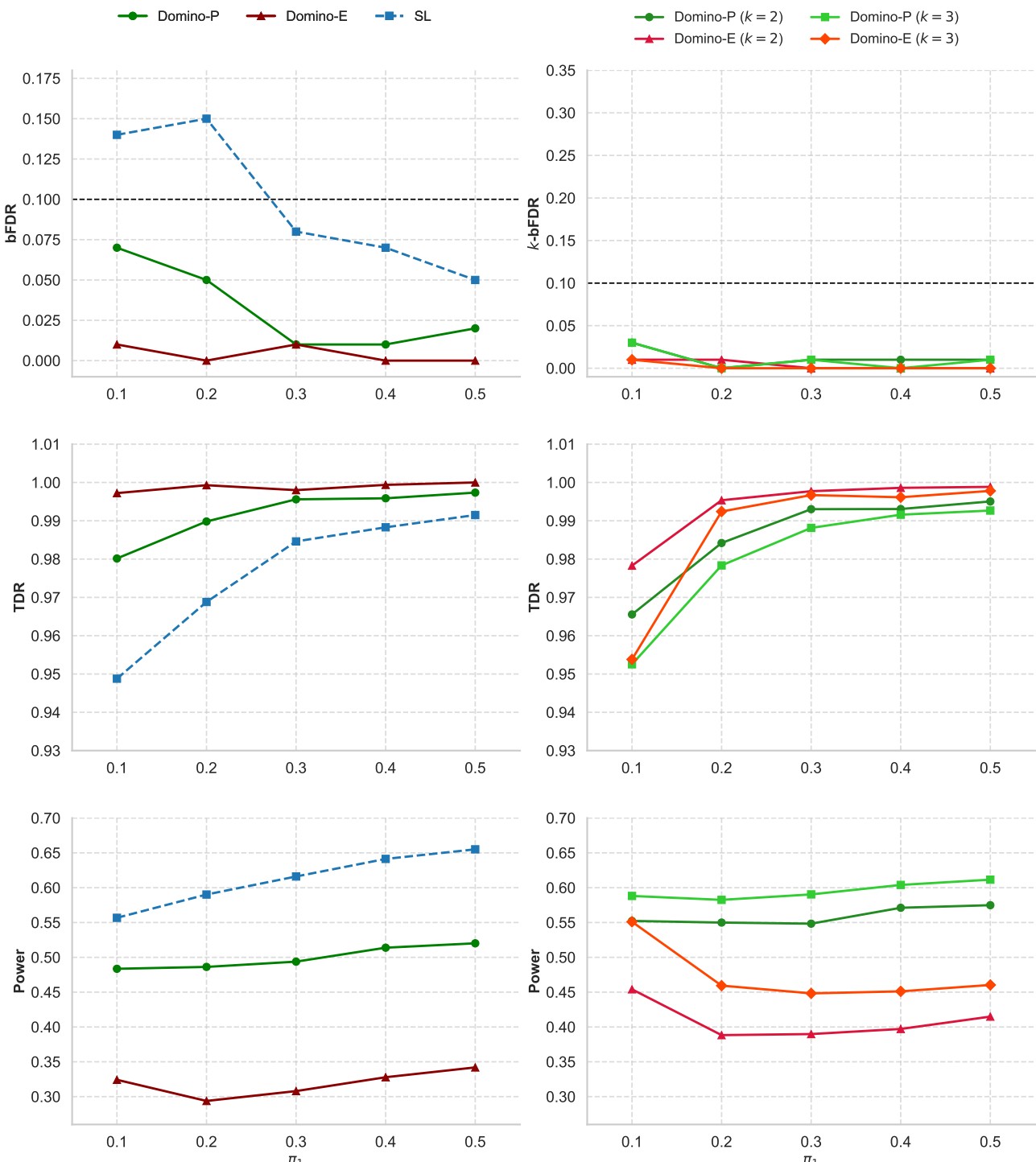

*Figure 4.* Performance comparison of Domino and SL methods across varying signal proportions $\pi_1$.

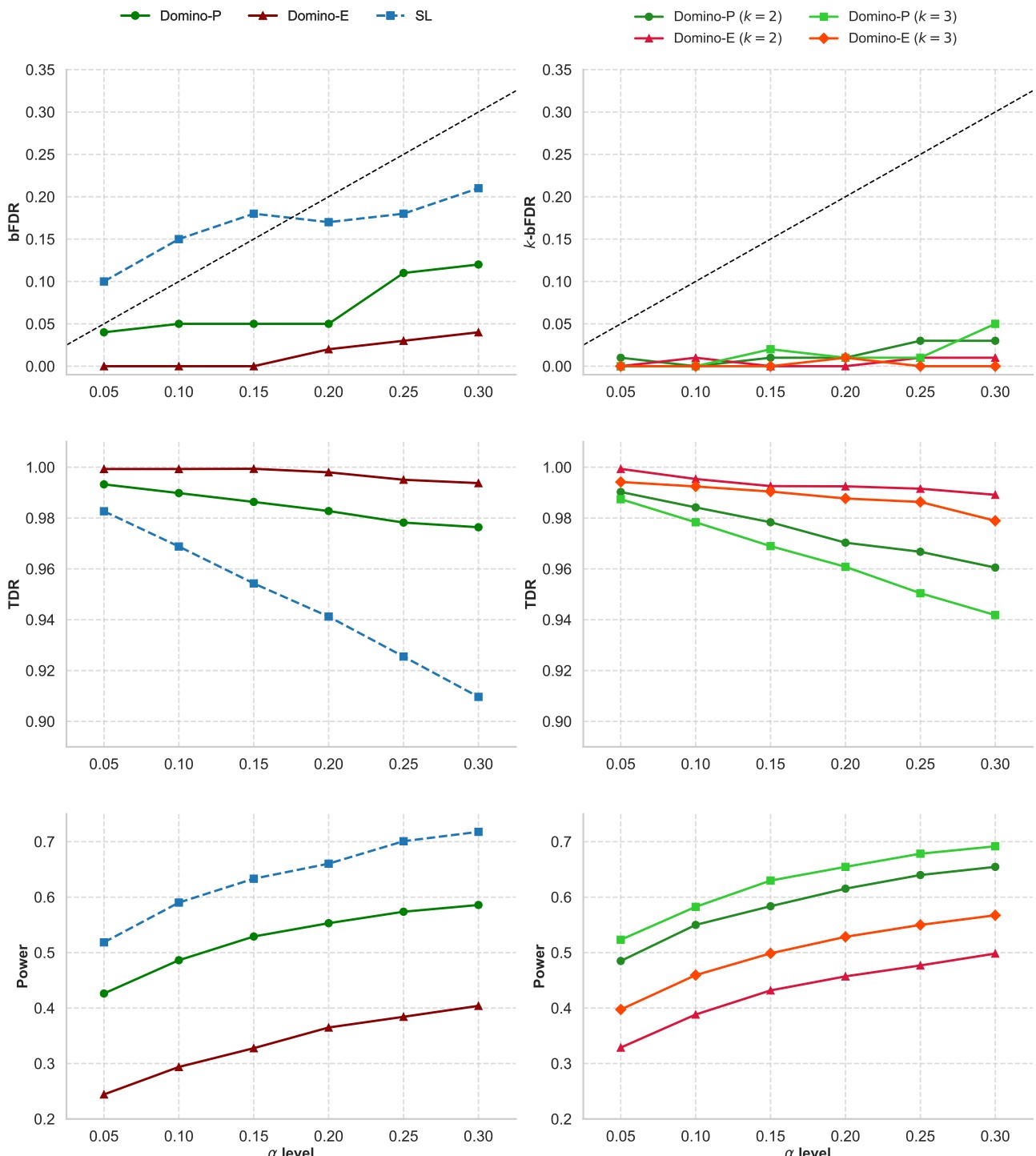

*Figure 5.* Performance comparison of Domino and SL methods across varying nominal levels $\alpha$.

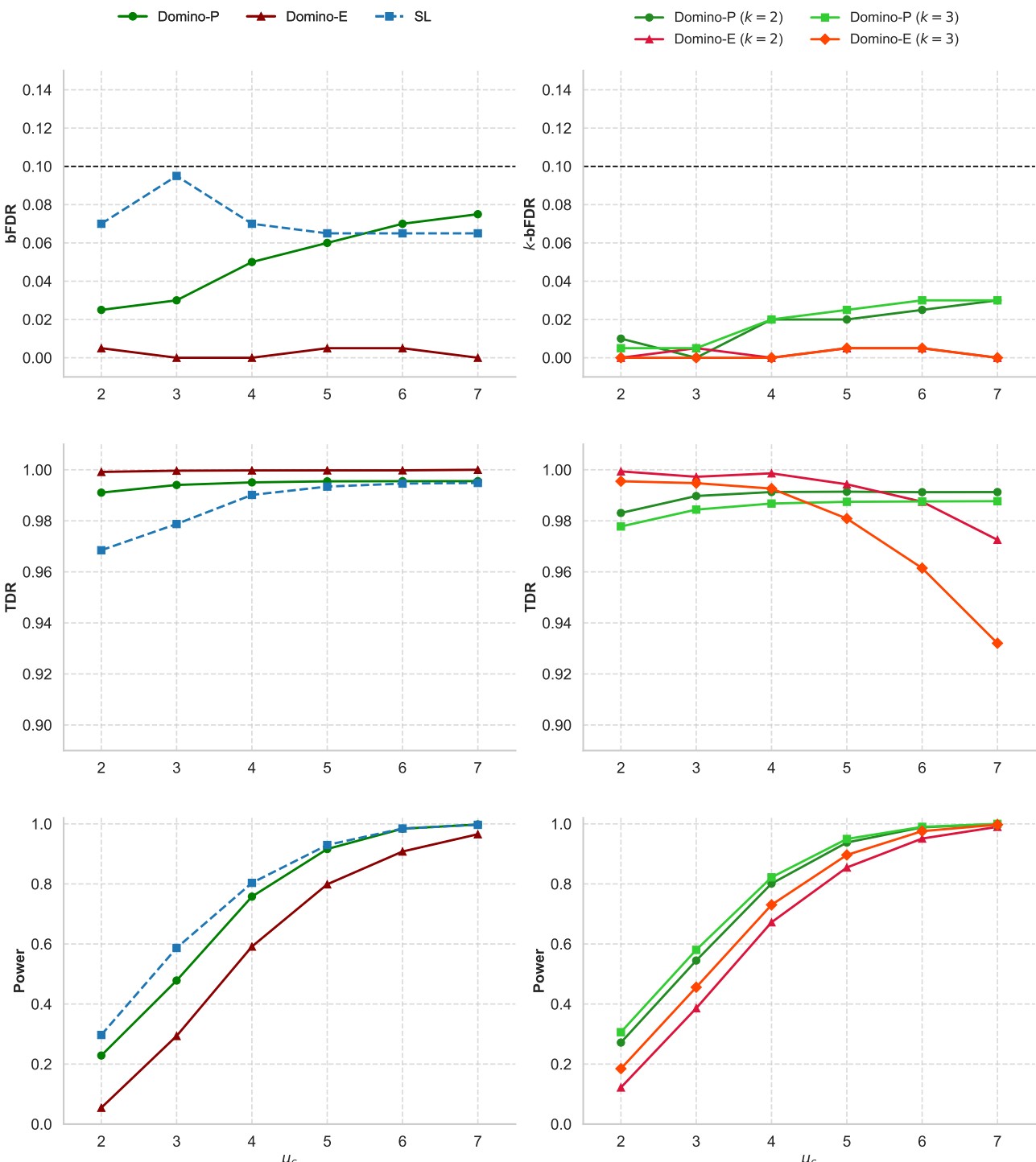

*Figure 6.* Performance comparison of Domino and SL methods across varying signal strengths $\mu_c$.

