# OpenReview forum: "Generalized Boundary FDR Control under Arbitrary Dependence: An Approach on Closure Principle"
_ICML.cc/2026/Conference — ICML 2026 regular_

### Official Review · Reviewer_pznL · 2026-02-21

**Soundness:** 2
**Presentation:** 3
**Significance:** 3
**Originality:** 2
**Overall Recommendation:** 3
**Confidence:** 4

**Summary:**

This paper proposes the Domino framework to control the $k$-boundary False Discovery Rate ($k$-bFDR) under arbitrary dependence structures. The motivation is well-justified: standard FDR control often suffers from the "hitchhiking" effect, where strong signals mask marginal errors. By enforcing a strict boundary threshold via $k$-bFDR, the authors aim to provide a more robust solution for downstream experimental validation. Methodologically, the paper incorporates $k$-bFDR into the e-closure framework. Notably, it provides not only an e-value-based method but also a p-value-based approach (Domino-P), which mitigates the power loss caused by p-to-e calibrations.

**Compliance With Llm Reviewing Policy:**

Affirmed.

**Final Justification:**

I recommend a weak reject, but I am open to acceptance. The theoretical work is sound. The main contribution is the Domino-P method for bFDR control. The existing e-closure framework is broad and solves many related problems. This prior work limits the significance of the current paper. Furthermore, the power of the bFDR control algorithm is limited. I appreciate the strong soundness, but I have reservations regarding the practical significance.

**Key Questions For Authors:**

1. Could you thoroughly explain the mathematically contradictory results in Figure 5 and Figure 6? Specifically, why does the power decrease as the nominal level $\alpha$ increases, and why does it decrease as the signal strength $\mu_c$ increases?

2. Given that the current extreme signal strength causes a ceiling effect (where all methods trivially approach a power of 1), can you provide new simulation results under a moderate signal strength regime?

3. In the paragraph "Comparing bFDR and FDR," the paper argues that bFDR is advantageous because it imposes a fixed threshold (e.g., $1/\alpha$), contrasting this with FDR's control over "average quality." However, this argument appears somewhat unconvincing when viewed through the lens of the e-closure paper. Recall that the standard e-BH procedure controls FDR by requiring the ordered e-values to meet a strict threshold condition (i.e., rejecting up to $k = \max\lbrace i: e_{(i)} \ge m/(i\alpha)\rbrace$. The $\overline{\text{e-BH}}$ procedure proposed in the e-closure framework improves upon standard e-BH precisely by relaxing this strict per-element constraint and leveraging the arithmetic average of e-values across subsets to gain power. Does this seem to be exactly the opposite of bFDR's idea?

4. The e-closure framework establishes a universal rule: once an error rate function $f$ is given, defining the rejection set as $\mathcal{R}_{\alpha}^{\text{ER}_f}(\mathbf{E}) := \lbrace R \in 2^{[m]} : \mathbf{e}_S \ge \frac{f_S(R)}{\alpha} \quad \forall S \in 2^{[m]} \rbrace$ immediately provides a procedure that controls the expectation $\mathbb{E}[f_N(R)] \le \alpha$. In Proposition 4.1, this paper defines the specific error rate function $f_S(R)$ for $k$-bFDR. Substituting this definition into the existing e-closure framework directly yields a valid method for $k$-bFDR control. Given this straightforward "plug-and-play" derivation for the e-value procedure, what should readers consider the most prominent and distinct theoretical contribution of this paper?

**Limitations:**

yes

**Strengths And Weaknesses:**

Strengths:

* Well-motivated Problem Setting: The authors tackle a highly relevant and well-recognized challenge in the realm of multiple testing: the "hitchhiking" effect, wherein overwhelmingly strong signals can inadvertently mask marginal or erroneous discoveries under standard False Discovery Rate (FDR) control. By generalizing the traditional boundary FDR to the $k$-bFDR metric, the paper offers a much more nuanced and theoretically sound approach to handling this issue.

* P-value Compatibility:The development and inclusion of the Domino-P procedure stand out as a highly commendable and practically significant contribution of this work. By demonstrating how to seamlessly embed p-value statistics (e.g., generalized Bonferroni, Simes) into the framework, the authors provide a practical tool that avoids the power loss typically associated with p-to-e calibrations.

Weaknesses:

* Suspicious Error in Empirical Results: The results presented in Figure 5 and 6 mathematically contradict the fundamental properties of hypothesis testing.Figure 5 (Power vs. $\alpha$): As the nominal level $\alpha$ increases, the power is expected to monotonically increase or remain stable. However, Figure 5 shows the power mysteriously decreasing as $\alpha$ increases.Figure 6 (Power vs. $\mu_c$): As the signal strength $\mu_c$ increases, the statistical evidence becomes stronger, which should lead to a monotonically increasing power. Yet, Figure 6 shows the power decreasing as the signal strengthens.These phenomena may indicate some bugs in the underlying simulation code.

* Uninformative Simulations Due to Over Strong Signal: In the simulations, it appears that the empirical power for almost all methods is very close to 1. This might originate from an overly strong signal setting. For example, in the Section 5.1 simulation, the exact value of $\mu_c$ is not explicitly specified in the main text, but referencing Appendix D, I deduce it to be $\mu_c = 6$. It raises the question: is this signal too strong? While this overall data generation setting is quite common in the multiple testing literature (e.g., Asynchronous Online Testing of Multiple Hypotheses; SAFFRON: an adaptive algorithm for online control of the false discovery rate), previous works generally do not adopt such high signal strengths, often preferring a more moderate regime like $\mu_c \in \lbrace 2, 3 \rbrace$. Evaluating the proposed method under weaker signals would likely better illustrate its actual empirical advantages and reveal true efficiency differences between the methods.

* Over-claiming of Theoretical Contributions: The theoretical narrative surrounding the e-value formulation is somewhat overstated. First, extending the target metric from bFDR to $k$-bFDR is merely a definitional modification. More importantly, from an e-value perspective, developing a procedure to control this new metric under arbitrary dependence does not constitute a fundamental methodological breakthrough. As established by the e-closure framework, one can control any error rate expressible as an expectation simply by substituting the corresponding loss function. Thus, the Domino-E procedure is essentially a direct instantiation of the e-closure principle with a specific loss function.

---

> ### Author Rebuttal · Authors · 2026-03-31
>
> Thanks for your comments on our paper. We would like to part-wisely respond to your comments.
>
> > W1&Q1: Suspicious Error in Empirical Results.
>
> We apologize for the confusion. We clarify that the TDR in Figures 5 & 6 (and elsewhere) is defined as the expected proportion of true positives among rejections (Lines 281-282, right column). Note that TDR does not measure power but rather reflects the quality of rejections, which is consistent with k-bFDR control. Thus, as $\alpha$ or $\mu_c$ increases, the method tends to produce a larger rejection set, leading to a decrease in TDR.
>
> > W2&Q2: Uninformative Simulations Due to Over Strong Signal.
>
> Thanks for the suggestion. Following the settings in the paper, we conduct simulations under moderate signal strengths $\mu_c\in ${2,3} to compare Domino with SL across varying correlations and proportions of non-nulls. Per **Reviewer hCXq's Q3**, we adopt a truncated normal distribution to generate $\mu_j$.
>
> As shown in Tables 4-5 (and Table 3 in **our response to Reviewer hCXq**), Domino yields a rejection set with higher TDR, whereas SL exhibits bFDR inflation. We emphasize TDR here to prioritize rejection quality under bFDR control instead of pursuing the number of rejections.
>
> Table 4: The results of different methods under different correlations $\rho$ with $\pi_1=0.2,\alpha=$5%.
>
> |||bFDR (%)|||TDR (%)|||Rej.|||
> |-|-|-|-|-|-|-|-|-|-|-|
> |$\mu_c$|$\rho$|Domino-P|Domino-E|SL|Domino-P|Domino-E|SL|Domino-P|Domino-E|SL|
> |2|-1/99|2|0|6|99.3|100|97.9|3.8|0.6|4.9|
> ||0|2|0|5|98.9|99.5|98.1|3.9|0.5|5.1|
> ||0.25|2|0|3|99.4|100|98.8|3.7|0.6|4.9|
> ||0.9|1|0|2|99.5|100|98.9|4.2|0.7|5.9|
> |3|-1/99|2|0|4|99.6|100|98.8|8.4|4.9|10.3|
> ||0|2|0|5|99.5|99.9|98.7|8.4|5.0|10.4|
> ||0.25|2|0|6|99.6|100|98.9|8.4|4.8|10.3|
> ||0.9|1|0|2|99.5|100|98.9|8.7|4.9|11.2|
>
> Table 5: The results of different methods across varying signal proportions $\pi_1$ with $\rho=0.5,\alpha=$5%.
>
> |||bFDR (%)|||TDR (%)|||Rej.|||
> |-|-|-|-|-|-|-|-|-|-|-|
> |$\mu_c$|$\pi_1$|Domino-P|Domino-E|SL|Domino-P|Domino-E|SL|Domino-P|Domino-E|SL|
> |2|0.1|3|0|7|98.9|100|96.2|2.1|0.5|3.3|
> ||0.2|4|0|5|99.5|100|98.7|4.1|0.7|5.3|
> ||0.3|1|0|2|99.7|100|99.1|5.5|0.8|8.2|
> ||0.4|0|0|3|99.9|100|99.2|8.1|1.5|12.5|
> ||0.5|2|0|6|99.7|100|99.2|10.6|2.1|16.9|
> |3|0.1|3|1|8|99.1|99.8|96.5|4.5|2.5|6.1|
> ||0.2|4|0|5|99.6|100|98.8|8.6|5.0|10.8|
> ||0.3|1|1|4|99.8|99.9|99.3|12.8|7.2|15.6|
> ||0.4|1|0|4|99.9|100|99.7|18.2|10.9|23.2|
> ||0.5|2|1|3|99.7|100|99.5|23.3|14.6|29.4|
>
>
> > W3: Over-claiming of Theoretical Contributions.
>
> We would like to clarify the scope of our contribution:
>
> + One of our contributions is **Domino** itself, which establishes k-bFDR control under arbitrary dependence via closure principle. The main text focuses on using p-values; the e-value variant is discussed in Appendix B. This framework is **not** derived from the e-closure principle, nor does it involve p-to-e calibrations.
> + For e-values, we explicitly relate Domino-E to the e-closure principle (Lines 284-305, left column). As shown in Proposition 4.1, the e-closure principle for k-bFDR control is a special case of Domino-E under a specific k-local test.
> + Furthermore, Domino offers a unified framework accommodating different k-local tests. Thus, it is anticipated that integrating more powerful k-local tests into Domino-E would outperform the e-closure principle with k-bFDR control.
>
> > Q3: In..."Comparing bFDR and FDR"...Does this seem to be exactly the opposite of bFDR's idea?
>
> Apologies for the confusion caused. This paragraph (Lines 306-329, left column) provides a purely conceptual comparison of bFDR and FDR **by analyzing the rejection process**: bFDR control imposes a **fixed** threshold **only** on the least significant rejection, whereas FDR control applies an **adaptive** threshold to **every** rejection. These differing targets and thresholds highlight the distinct emphases of the two metrics: bFDR governs _marginal quality_, while FDR governs _average quality_.
>
> We will revise the paragraph and rename it to "Discussion of bFDR & FDR" for clarity.
>
> > Q4: ...what...the most prominent and distinct theoretical contribution of this paper?
>
> + We first clarify the methodological and theoretical contributions of our paper. Methodologically, we **develop Domino**, a unified framework that applies _closure principle_ to achieve valid k-bFDR control compatible with _both p-values and e-values_, allowing flexible choices of k-local test. Theoretically, we prove in Theorem 3.4 that Domino guarantees **k-bFDR control under arbitrary dependence**.
> + We appreciate that e-closure principle is a general framework for error rate control using e-values. As discussed in **our response to W3**, its construction for k-bFDR control is a special case of Domino-E under a specific k-local test. Proposition 4.1 introduces the error rate function $f_S(R)$ to facilitate the analysis of our method relative to e-closure principle; it is not the foundation from which Domino is derived.

---

> > ### Author Rebuttal · Reviewer_pznL · 2026-04-03
> >
> > Thank you for providing the additional results under moderate signal strengths ($\mu_c \in \{2, 3\}$). I would like to follow up on the first point, as these new settings make it even more critical to compare Power rather than TDR.
> >
> > By definition, TDR is simply 1 - FDR. Your algorithm is designed to control bFDR, and the final metric compared is 1 - FDR. It is actually quite normal that the result approaches 1. I think even if it were replaced with 1 minus other common error rates, the results would not be bad either. You mentioned in the rebuttal that under moderate signals, Domino "sacrifices a certain amount of rejections" to maintain this high TDR. This is precisely my concern: an overly conservative method that only rejects a tiny fraction of true signals will still boast a near-perfect TDR, completely masking the potentially severe drop in Power.
> >
> > Given the scarcity of existing methods designed for bFDR control, readers will naturally understand the inherent trade-offs involved. Therefore, explicitly demonstrating Power would not undermine your method's contribution; rather, it would provide a transparent and much-needed perspective on the practical cost of strict boundary control.

---

> > > ### Author Response · Authors · 2026-04-04
> > >
> > > Thank you for the constructive comment and your valuable suggestion! We would like to make the following clarification and supplement the Power-related results in the revision to provide a more comprehensive evaluation of our method's performance.
> > >
> > > + We initially intended to illustrate the *trade-off* between controlling bFDR to achieve a high-quality rejection set and the ability to detect true signals by *reporting the number of rejections*, since the number of rejections and Power exhibit similar trends. Meanwhile, we show the TDR, which *reflects the quality of the rejection set*, to evaluate the effectiveness of different methods. Relevant results have been provided in the paper and Tables 3-5 in the rebuttal. These results indicate that the pursuit of a high-quality set via bFDR control tends to produce a modest discovery count, consequently diminishing Power.
> > > + To help readers better understand the inherent trade-offs involved, we will supplement the results in the revised version with an explicit comparison of Power across different methods, presented alongside the TDR results. This will provide an intuitive illustration of the discovery capability achieved by our method under rigorous bFDR control.
> > > + The Power-supplemented results are illustrated in Tables 6-9, each corresponding to the experimental scenarios in the rebuttal, as indicated in the respective titles. Compared to p-value-based methods, Domino-E exhibits a lower Power, which may be attributed to the characteristic of e-values. Though the Power of Domino-p is slightly lower than SL, SL suffers a bFDR inflation under dependence.
> > >
> > > Table 6: The results of different methods on the gene data study. (Consistent with Table 2, Power results have been added.)
> > >
> > > |  | bFDR (%) | | | | TDR (%) | | | | Power (%) | | | |
> > > | --- | --- | --- | --- | --- | --- | --- | --- | --- | --- | --- | --- | --- |
> > > | $\alpha$ (%) | Domino-P | Domino-E | SL | BH | Domino-P | Domino-E | SL | BH | Domino-P | Domino-E | SL | BH |
> > > | 5 | 4 | 2 | 19 | 98 | 99.4 | 99.4 | 98.8 | 96.8 | 86.5 | 84.9 | 90.4 | 93.7 |
> > > | 10 | 6 | 2 | 27 | 100 | 99.3 | 99.4 | 98.7 | 95.8 | 87.2 | 85.5 | 90.9 | 94.8 |
> > > | 20 | 6 | 7 | 32 | 100 | 99.3 | 99.4 | 98.5 | 92.4 | 87.9 | 86.1 | 91.5 | 96.2 |
> > >
> > > Table 7: The results of different Domino methods with $\alpha= $5%, $\pi_1=0.2 $. (Consistent with Table 3, Power results have been added.)
> > >
> > > |  |  | $ \rho=0 $ | | | $ \rho = 0.25 $ | | |
> > > | --- | --- | --- | --- | --- | --- | --- | --- |
> > > |  | Local test | k-bFDR (%) | Power (%) | Rej. | k-bFDR (%) | Power (%) | Rej. |
> > > | $ \mu_c = 2 $ | 1,Simes | 2 | 19.7 | 3.9 | 2 | 19.2 | 3.7 |
> > > | | 1,Harmo | 0 | 9.2 | 1.8 | 0 | 9.2 | 1.8 |
> > > | | 1,eAVG | 0 | 2.6 | 0.5 | 0 | 3.2 | 0.7 |
> > > | | 2,Bonf | 0 | 23.5 | 4.8 | 1 | 23.0 | 4.6 |
> > > | | 2,e-closure | 0 | 8.8 | 1.7 | 0 | 9.4 | 1.8 |
> > > | | 3,Bonf | 0 | 26.3 | 5.3 | 0 | 25.9 | 5.2 |
> > > | | 3,e-closure | 0 | 14.5 | 2.8 | 0 | 15.2 | 3.0 |
> > > | $ \mu_c = 3 $ | 1,Simes | 0 | 42.7 | 9.6 | 1 | 41.9 | 9.5 |
> > > | | 1,Harmo | 0 | 27.0 | 1.8 | 0 | 26.5 | 1.9 |
> > > | | 1,eAVG | 0 | 25.0 | 0.5 | 0 | 24.4 | 0.6 |
> > > | | 2,Bonf | 0 | 48.4 | 9.6 | 1 | 47.6 | 9.5 |
> > > | | 2,e-closure | 0 | 33.9 | 6.7 | 0 | 32.9 | 6.5 |
> > > | | 3,Bonf | 0 | 51.9 | 10.3 | 0 | 51.5 | 10.3 |
> > > | | 3,e-closure | 0 | 40.6 | 8.0 | 0 | 39.3 | 7.8 |
> > >
> > >
> > > Table 8: The results of different methods under different correlations $\rho$ with $\pi_1=0.2, \alpha=$5%. (Consistent with Table 4, Power results have been added.)
> > >
> > > |  |  | bFDR (%) | | | Power (%) | | | Rej. | | |
> > > |-|-|-|-|-|-|-|-|-|-|-|
> > > | $ \mu_c $ | $ \rho $ | Domino-P | Domino-E | SL | Domino-P | Domino-E | SL | Domino-P | Domino-E | SL |
> > > | 2 | -1/99 | 2 | 0 | 6 | 19.1 | 2.7 | 24.0 | 3.8 | 0.6 | 4.9 |
> > > | | 0.0 | 2 | 0 | 5 | 19.7 | 2.6 | 24.8 | 3.9 | 0.5 | 5.1 |
> > > | | 0.25 | 2 | 0 | 3 | 18.6 | 3.1 | 24.0 | 3.7 | 0.6 | 4.9 |
> > > | | 0.9 | 1 | 0 | 2 | 19.2 | 3.2 | 25.3 | 4.2 | 0.7 | 5.9 |
> > > | 3 | -1/99 | 2 | 0 | 4 | 42.0 | 24.5 | 50.7 | 8.4 | 4.9 | 10.3 |
> > > | | 0.0 | 2 | 0 | 5 | 42.8 | 25.0 | 51.8 | 8.4 | 5.0 | 10.4 |
> > > | | 0.25 | 2 | 0 | 6 | 42.3 | 24.2 | 51.2 | 8.4 | 4.8 | 10.3 |
> > > | | 0.9 | 1 | 0 | 2 | 41.9 | 24.4 | 52.5 | 8.7 | 4.9 | 11.2 |
> > >
> > >
> > > Table 9: The results of different methods across varying signal proportions $\pi_1$ with $\rho=0.5,\alpha=$5%. (Consistent with Table 5, Power results have been added.)
> > >
> > > |  |  | bFDR (%) | | | Power (%) | | |Rej. | | |
> > > |-|-|-|-|-|-|-|-|-|-|-|
> > > | $\mu_c$ | $\pi_1$ | Domino-P | Domino-E | SL | Domino-P | Domino-E | SL | Domino-P | Domino-E | SL |
> > > |2|0.1|3|0|7|19.2|4.5|22.6|2.1|0.5|3.3|
> > > | |0.2|4|0|5|20.0|3.4|24.6|4.1|0.7|5.3|
> > > | |0.3|1|0|2|18.8|2.8|27.7|5.5|0.8|8.2|
> > > | |0.4|0|0|3|20.7|3.9|31.1|8.1|1.5|12.5|
> > > | |0.5|2|0|6|21.1|4.2|33.2|10.6|2.1|16.9|
> > > | 3 | 0.1 | 3 | 1 | 8 | 42.7 | 24.0 | 49.1 | 4.5 | 2.5 | 6.1 |
> > > | | 0.2 | 4 | 0 | 5 | 43.1 | 24.9 | 52.6 | 8.6 | 5.0 | 10.8 |
> > > | | 0.3 | 1 | 1 | 4 | 43.9 | 24.8 | 52.7 | 12.8 | 7.2 | 15.6 |
> > > | | 0.4 | 1 | 0 | 4 | 46.6 | 27.9 | 59.1 | 18.2 | 10.9 | 23.2 |
> > > | | 0.5 | 2 | 1 | 3 | 46.6 | 29.4 | 58.9 | 23.3 | 14.6 | 29.4 |

---

### Official Review · Reviewer_U2UR · 2026-02-26

**Soundness:** 4
**Presentation:** 4
**Significance:** 4
**Originality:** 4
**Overall Recommendation:** 5
**Confidence:** 4

**Summary:**

This paper addresses a critical gap in multiple hypothesis testing: the lack of reliability guarantees for marginal discoveries at the boundary of rejection sets.  The authors propose k-bFDR, a generalization of the recently introduced bFDR that controls the joint error probability of the k least significant discoveries. Building on the closure principle, they develop Domino, a unified framework achieving k-bFDR control under arbitrary dependence for both p-values and e-values.

**Compliance With Llm Reviewing Policy:**

Affirmed.

**Key Questions For Authors:**

1. For practitioners applying Domino, what principles should guide the choice of k?

**Limitations:**

yes.

**Strengths And Weaknesses:**

Strength:
1. The paper is technically rigorous throughout. The theoretical foundation rests on well-established closure principles, extended thoughtfully to boundary control.
2. The paper is well-structured with clear logical flow.
3. The paper addresses a genuinely important problem.
4.  While closure testing and bFDR are not new, their combination yields genuine novelty.
Weakness:
The paper could more explicitly discuss when practitioners should prefer k-bFDR over FDR or FWER and practical guidance on selecting k remains somewhat implicit.

---

> ### Author Rebuttal · Authors · 2026-03-31
>
> Thanks for your comments on our paper. We would like to part-wisely respond to your comments.
>
> > W1: The paper could more explicitly discuss when practitioners should prefer k-bFDR over FDR or FWER and practical guidance on selecting k remains somewhat implicit.
> >
> > Q1: For practitioners applying Domino, what principles should guide the choice of k?
>
> Thank you for your question! This is fundamentally a practice-oriented question. Choosing among k-bFDR, FDR, and FWER, along with selecting an appropriate value for $k$, is not about determining which criterion is inherently better. Instead, it depends on the specific risk preferences and decision-making context involved. From a decision-theoretic perspective, these criteria reflect different risk priorities. Selecting an inappropriate criterion may lead to results that lack practical significance.
>
> **When to prefer k-bFDR:**
>
> In contrast to FDR, which controls the _average_ error rate, and FWER, which guards against _any_ error, k-bFDR provides a targeted guarantee on the reliability of the $k$ boundary (weakest) discoveries, which are often the most decision-critical in practice. This makes it particularly well-suited in settings where decisions are based on a ranked list with a cutoff, and where only a limited number of top findings are acted upon. For example, selecting a small set of genes for costly wet-lab validation, detecting anomalies in factory quality inspection, or targeting high-value customers in precision marketing. In these settings, the primary risk is not the average error but that the last few admitted discoveries are spurious; k-bFDR directly addresses this gap.
>
> **How to choose $k$:**
>
> + Regarding the choice of $k$, it should be guided by the intended use of the selected set and the practitioner's tolerance for boundary risk. More generally, $k$ can be interpreted as the size of the "critical frontier" within the rejection set for which reliability is essential, and should therefore align with resource constraints or downstream validation capacity. For instance, if only $k$ candidates can be validated or deployed, setting $k$ accordingly ensures that the entire actionable subset enjoys a rigorous boundary guarantee. This interpretation makes k-bFDR a natural middle ground between global (FDR) and worst-case (FWER) control, offering a principled and application-aligned way to trade off power and reliability.
> + If one has a downstream task for which the choice of $k$ may lead to varying performance, then a data-driven approach to selecting $k$ would be desirable. However, this problem is highly challenging, as it introduces issues of double-dipping. Developing a method that enables the selection of $k$ with valid theoretical properties would require a carefully redesigned framework, especially given that different downstream tasks may prioritize different performance metrics. As such, this remains an important direction for future research.
>
> We will incorporate these discussions into the revision to make the practical implications of k-bFDR and the choice of $k$ more explicit.

---

> > ### Author Rebuttal · Reviewer_U2UR · 2026-04-02
> >
> > Thank you for the reply. I maintain my accept rating

---

> > > ### Author Response · Authors · 2026-04-02
> > >
> > > Thank you for your insightful comments and for your engagement in reviewing our manuscript. We will incorporate these discussions into the revision to enrich the content of our paper. Thank you once again for your support of our work!

---

### Official Review · Reviewer_hCXq · 2026-03-06

**Soundness:** 4
**Presentation:** 3
**Significance:** 4
**Originality:** 4
**Overall Recommendation:** 6
**Confidence:** 5

**Summary:**

FDR (false discovery rate) is the expected value of FDP (false discovery proportion). The majority of existing multiple testing procedures control FDR, but we must observe that these only control the FDP *in expectation* (as opposed to controlling a *probability*). This allows for the following pathology to enter: extremely significant results skew the distribution of FDP, allowing for weak, possibly false positive, signals to enter the rejection set.

To counter this deficiency, previous authors (Soloff et al., 2024; Xiang et al., 2025) propose the boundary false discovery
rate (bFDR), defined as the probability that the last rejected hypothesis is a false discovery. Prior art is, however, limited to the specific case on independent p-values.

The authors first generalize the definition to $k$-bFDR, the probability of falsely rejecting the hypotheses corresponding to the “k least significant” discoveries in R. bFDR is then achieved with the special case $1$-bFDR.

A primary contribution is that this paper relaxes the independence assumptions required by prior art. The authors first define a "Domino" condition in definition 3.3, and prove in Theorem 3.4 that their domino algorithm controls $k$-bFDR under *arbitrary* dependence, broadening the scope of problems where bFDR can be controlled.

The domino algorithm is somewhat "meta" or "modular" in that the user has freedom to choose the $k$-local test. It therefore supports both $p$-value and $e$-value testing procedures.

Does the procedure sacrifice anything in order to relax the independence assumption? The simulated data suggest that TDR (true discovery rate) is higher than the incumbent *support line* procedure of Soloff et al., 2024, while controlling $k$-FDR. This also appears to be true for the bootstrapped genomics simulations.

**Compliance With Llm Reviewing Policy:**

Affirmed.

**Final Justification:**

This paper is a well written and, above all, timely contribution to the fast growing literature on multiple testing with e-values. The paper is well written, sound, and has scope to improve FDR controlling procedures which are heavily used.

**Key Questions For Authors:**

Why did you use an oracle e-variable for demonstrating Domino-E and why was it dropped from the genetics example? Is it because e-variables for unknown/composite alternative, such as mixtures, were not powerful enough for your illustrations to be compelling?

It's generally the case that e-variables are able to handle arbitrary dependence in multiple testing correction algorithms. Yet your domino-P also controls k-bFDR under arbitrary dependence. In what cases then would domino-E ever be favorable over domino-P?

In your numerical simulation you generate means under the alternative (>0) from a Gaussian. I realize the mean is many standard deviations away from zero, but you could technically get a negative with low probability. Maybe generate from a truncated Gaussian instead?

**Limitations:**

One might expect that one would have to give something up (in terms of performance) in order to gain robustness to arbitrary dependence, but I don't see this discussion in the paper. The simulation studies suggest we have both improved performance AND robustness to dependence, but these are only two simulation studies. Can the authors be more transparent about the tradeoffs, and a discussion perhaps of where this methodology would not be appropriate?

I believe the authors are honest about whether it makes sense to focus on the $k$ weakest discoveries at the boundary. This is motivated by the monotonicity assumption discussed in section 2.2: "When monotonicity is violated, the rejection at the boundary characterized by bFDR may not necessarily correspond to the theoretically least compelling scenario." In practical settings it is not clear whether this monotonicity assumption holds.

**Strengths And Weaknesses:**

Soundness: The theoretical contributions are sound. I find the numerical comparison with Domino-E a little disappointing as it uses an oracle mean in the alternative $\mu_c$, whereas in most applied work this would not be known. Domino-E also disappears from the empirical study on gene data. This isn't a big setback for the paper, as Domino-P is present in both, which is one of the novel contributions of the paper anyway.

Presentation: Presentation is good but I found myself wondering how this "hitchhiking" phenomenon actually occurs mechanically. The mathematical explanation of this hitchhiking isn't presented until the end of page 6, where we can see the comparison to a fixed and adaptive threshold, and how strong signals increase the denominator but do nothing to the numerator, reducing the threshold. I suggest the authors to bring this forward in the paper so readers have a clear understanding early on for how the hitchhiking happens.

Significance: Very significant contributions as relaxing independent to arbitrary dependence broadens the scope of the procedure.

Originality: The authors did not make the fundamental observation of hitchhiking and bFDR. This is attributed to prior art. The authors, however, extend the definition and provide a novel domino algorithm for controlling k-bFDR under arbitrary dependence. Their algorithm uses pre-existing k-local tests, from both p-value and e-value schools.

---

> ### Author Rebuttal · Authors · 2026-03-31
>
> Thanks for your comments on our paper. We would like to part-wisely respond to your comments.
>
> > W1&Q1: Why did you use an oracle e-variable...why was it dropped...?
>
> We would like to make the following clarification.
>
> + In our experiments, the true signal $\mu_j$ for alternatives was drawn from a normal distribution with mean $\mu_c$ (later replaced with a truncated normal per **your Q3**). We used $\mu_c$ as a _crude proxy_ for the oracle $\mu_j$ to construct e-values instead of using $\mu_j$ itself. This aligns with existing work using likelihood ratios as e-values, which also requires specifying the signal under the alternative (Wang & Ramdas, 2022).
> + We apologize for initially omitting Domino-E from gene data study, as we focused on p-value-based method comparisons. We have supplemented the results in Table 2. Compared to p-value-based methods, Domino-E yields a smaller but more precise rejection set yet still achieves valid bFDR control and high TDR.
>
> Table 2: The results of Domino-E on gene data study.
>
> |$\alpha$ (%)|bFDR (%)|TDR (%)|
> |-|-|-|
> |5|2|99.4|
> |10|2|99.4|
> |20|7|99.4|
>
> Ref:
>
> Wang, R., & Ramdas, A. (2022). False discovery rate control with e-values. Journal of the Royal Statistical Society Series B: Statistical Methodology, 84(3), 822-852.
>
> > W2: I suggest...how the hitchhiking happens.
>
> Thanks for your suggestion! We will explain the hitchhiking phenomenon earlier in the introduction for better comprehension.
>
> > Q2: ...In what cases then would domino-E ever be favorable over domino-P?
>
> Thanks for this insightful question. This may stem from the inherent properties of e-values versus p-values, as Domino-E and Domino-P frameworks are generally aligned.
>
> + The e-values, grounded in expectation, offer a natural combinability advantage over p-values to integrate into valid k-local tests or to yield k-local tests with higher power. For example, we construct a valid k-local test in the main text (Lines 244-254, left column). Moreover, Hartog & Lei (2025) showed weighted e-Bonferroni tests are more powerful than their p-value counterparts with the same weights and (1/e)-form p-values, indicating more efficient aggregation.
> + Domino-E is also preferable when e-values are easier to construct than p-values (e.g., complex, high-dimensional). For example, e-values can be formed by mixture likelihood ratios under composite nulls, avoiding the conservative calibration required for p-values (Zhang et al., 2024).
> + Furthermore, Table 3 (in **response to your Q3**) shows Domino-E is suitable when seeking a highly precise rejection set (few rejections, nearly all correct), as it yields fewer rejections than Domino-P but with TDR near one.
>
> We will incorporate this discussion into the final version.
>
> Refs:
>
> Hartog, W., & Lei, L. (2025). Family-wise error rate control with e-values. arXiv preprint arXiv:2501.09015.
>
> Zhang, Z., Ramdas, A., & Wang, R. (2024). On the existence of powerful p-values and e-values for composite hypotheses. The Annals of Statistics, 52(5), 2241-2267.
>
> > Q3: ...Maybe generate from a truncated Gaussian instead?
>
> Thanks for pointing this out! We will generate means from a truncated Gaussian and update the results accordingly.
>
> Table 3 shows Domino with different local tests. Performance trends align with Table 1 in Section 5.1. All methods achieve high TDR, confirming high-quality rejection sets.
>
> We also compare Domino with SL under varying correlations and non-null proportions in Tables 4-5 (in **response to Reviewer pznL's W2&Q2** due to space limitation).
>
> Table 3: The results of different Domino methods with $\alpha=$ 5%, $\pi_1=0.2$.
>
> |||$\rho=0$|||$\rho=0.25$|||
> |-|-|-|-|-|-|-|-|
> ||Local test|k-bFDR(%)|TDR(%)|Rej.|k-bFDR(%)|TDR(%)|Rej.|
> |$\mu_c=2$|1,Simes|2|99.4|3.9|2|89.1|3.7|
> ||1,Harmo|0|99.8|1.8|0|99.9|1.8|
> ||1,eAVG|0|99.5|0.5|0|100|0.7|
> ||2,Bonf|0|97.8|4.8|1|99.1|4.6|
> ||2,e-closure|0|99.7|1.7|0|100|1.8|
> ||3,Bonf|0|97.5|5.3|0|98.7|5.2|
> ||3,e-closure|0|98.9|2.8|0|99.7|3.0|
> |$\mu_c=3$|1,Simes|0|98.9|9.6|1|99.4|9.5|
> ||1,Harmo|0|99.8|1.8|0|99.9|1.9|
> ||1,eAVG|0|99.5|0.5|0|100|0.6|
> ||2,Bonf|0|98.9|9.6|1|99.4|9.5|
> ||2,e-closure|0|99.8|6.7|0|100|6.5|
> ||3,Bonf|0|98.9|10.3|0|99.2|10.3|
> ||3,e-closure|0|99.5|8.0|0|99.7|7.8|
>
> > Limitations:...Can the authors be more transparent about the tradeoffs...?
>
> We apologize for the absence of this limitation. Domino ensures valid k-bFDR control under arbitrary dependence; however, this generality may come at a cost of power, as it does not fully exploit the information under independence or certain dependence structures. We will add this limitation to the revision.
>
> We would like to clarify that TDR, defined as the expected proportion of true positives among rejections (Lines 281-282, right column), measures the quality of rejections, which is consistent with the aim of k-bFDR control. The TDR of Domino remains stable across dependence structures, likely due to the conservativeness of the rejection set.

---

> > ### Author Rebuttal · Reviewer_hCXq · 2026-04-01
> >
> > Thank you for the detailed response. As I had no earlier concerns with the paper, I will keep my rating as strong accept.

---

> > > ### Author Response · Authors · 2026-04-02
> > >
> > > We sincerely appreciate your helpful comments and strong support for our work! Your feedback has helped us improve the paper, and we will incorporate these clarifications into the revision. Thank you once again for your valuable suggestions and your engagement in reviewing our work.

---

### Official Review · Reviewer_vcK7 · 2026-03-12

**Soundness:** 3
**Presentation:** 3
**Significance:** 2
**Originality:** 3
**Overall Recommendation:** 3
**Confidence:** 4

**Summary:**

This paper studies boundary false discovery control in multiple testing. It argues that standard FDR control can still leave the least significant discoveries in the rejection set unreliable, which is problematic in applications where marginal discoveries matter. The paper introduces k-bFDR, a generalization of boundary FDR that controls the probability that the k least significant discoveries are all false. To control this criterion under arbitrary dependence, the paper proposes a framework called Domino, based on the closure principle, and develops versions applicable to both p-values and e-values. The paper also investigates the relationship between k-bFDR and existing error criteria, provides theoretical validity results for Domino, and includes simulations and real-data analyses to illustrate the method.

**Compliance With Llm Reviewing Policy:**

Affirmed.

**Final Justification:**

The rebuttal clarified several points and improved my view of the paper, although some concerns remain. Overall, I maintain my current assessment.

**Key Questions For Authors:**

- Q1. Can the authors clarify the practical motivation for boundary control, and explain more explicitly in what settings it is essential relative to existing false discovery criteria?

- Q2: Can the authors provide a clearer comparison with prior work, especially Soloff et al. (2024) and Xiang et al. (2025), to better highlight the novelty of the proposed framework and the contribution of Theorem 3.4?

- Q3: Can the authors comment on whether additional real-data evidence or ablation-style empirical comparisons could further demonstrate the practical gains of the method?

- Q4: Can the authors improve the exposition beginning in Section 2.1 so that notation is clearly introduced before being used?
- Q5: Where is the arbitrary dependence assumption reflected in the theory or proofs?
- Q6: At the end of Section 1, the paper mentions p-values and e-values. How are these incorporated into the later development of the paper?

**Limitations:**

No. The paper would benefit from a more explicit discussion of its limitations, including the assumptions under which the method is expected to work well and settings where its performance may degrade.

**Strengths And Weaknesses:**

- The proposed closure-based framework appears to be fairly general.
- The paper’s motivation should be strengthened. In particular, it is not yet fully clear why this type of boundary control is essential in practice.
- Although the proof of Theorem 3.4 seems correct, its novelty relative to prior bFDR work is not fully clear. A more explicit comparison with Soloff et al. (2024) and Xiang et al. (2025) would be helpful.
- The experiments seem reasonable, but the real-data evidence is not fully convincing, and the empirical results do not yet clearly demonstrate the practical gains of the proposed method.
- From Section 2.1 onward, some notation is introduced before being properly defined, which affects readability.

---

> ### Author Rebuttal · Authors · 2026-03-31
>
> Thanks for your comments on our paper. We would like to part-wisely respond to your comments.
>
> > W1&Q1:...clarify the practical motivation...?
>
> As discussed in introduction (Lines 19-44, right column), erroneous decisions in high-precision, high-risk settings can be catastrophic. For example, in genetic selection, incorrect choices may disrupt downstream research (e.g., drug development), wasting resources and endangering safety. In venture capital, poor investments can cause major financial losses. In such cases, one should prioritize the reliability of individual discoveries, especially those with weak evidence at the rejection boundary, over the average quality of the full set. The former aligns with k-bFDR, which is the focus of this paper; the latter is what FDR provides.
>
> > W2&Q2:...provide a clearer comparison with prior work...?
>
> 1. We first **propose k-bFDR**, a novel and flexible error metric that extends the bFDR (Soloff et al., 2024; Xiang et al., 2025) from a single marginal false discovery to a batch of $k$.
> 2. We **develop Domino**, a unified framework based on the _closure principle_ for k-bFDR control, compatible with _both p-values and e-values_. In contrast, SL (Soloff et al., 2024) is built on **independent p-values** for bFDR control.
> 3. We prove (in Theorem 3.4) that Domino guarantees **k-bFDR control under arbitrary dependence**. Whereas SL primarily ensures bFDR control under independent p-values. Since our result is built on the closure principle, the proof techniques differ from those of SL.
>
> We have highlighted these differences in the main text, such as in the introduction (Lines 74-81, left column) and in the discussion of Domino's theoretical property (Lines 207-214, right column). We will emphasize them more clearly in the revision.
>
> > W3&Q3:...demonstrate the practical gains of the method?
>
> We add a real data analysis on the GSE15852 gene expression dataset to further demonstrate the practical gains of Domino. We adopt the pathway combined scores to evaluate discovery quality under different methods (Kuleshov et al., 2016). A high score effectively captures true positive signals while filtering out false positives.
>
> We report 6 well-studied breast cancer pathway combined scores for all methods in Table 1. Our method yields higher-quality discovery sets that precisely exclude noise while retaining true signals. Although BH and SL reject more hypotheses, their score results reflect excessive noise.
>
> Table 1: Comparison of combined scores across different procedures with $\alpha=$ 5%. Values in parentheses represent rejection numbers.
>
> ||Domino(k=1) (390)|Domino(k=2) (431)|SL (1025)|BH (1769)|
> |-|-|-|-|-|
> |PPAR SP|**302**|259|174|111|
> |AMPK SP|125|**131**|50|22|
> |FAD|97|**126**|104|79|
> |ECM-RI|26|49|45|**61**|
> |Lipolysis Regulation|**95**|81|46|43|
> |Insulin SP|**61**|51|24|10|
>
> We further provide an ablation-style analysis of Domino under varying local test forms (Table 3 in **Reviewer hCXq's Q3**), correlations and non-null proportions (Tables 4-5 in **Reviewer pznL's W2&Q2**). In fact, we have already conducted experiments with different settings in Appendix D.
>
> Ref:
>
> Kuleshov, M. V., Jones, M. R., Rouillard, A. D., Fernandez, N. F., Duan, Q., Wang, Z., ... & Ma'ayan, A. (2016). Enrichr: a comprehensive gene set enrichment analysis web server 2016 update. Nucleic acids research, 44(W1), W90-W97.
>
> > W4&Q4:...improve...notation...?
>
> Thank you for your advice! We will revise Sections 2.1 onward to ensure notation is properly introduced before use for clarity.
>
> > Q5:...arbitrary dependence assumption reflected...?
>
> In multiple testing, the _arbitrary dependence assumption_ actually means no conditions are imposed on the dependence structure of test statistics, i.e., **no additional assumptions are required**. Thus, our method doesn't need dependence assumptions with k-bFDR control, demonstrating generality and applicability under complex dependence in practice. In contrast, some classical procedures rely on the independence assumption, which is hard to verify and often violated in practice.
>
> > Q6: ...the paper mentions p-values and e-values...?
>
> In multiple testing, each null hypothesis is linked to a test statistic (e.g., p-value or e-value), and rejection occurs when statistic indicates significant evidence, e.g., a small p-value or large e-value. The statistic magnitude reflects significance, used in k-bFDR definition (Lines 153–159, left col). In Domino, p-values or e-values rank hypotheses by significance and enable valid k-local tests, as detailed in Sections 3.1 & 3.2.
>
> > Limitations: No...
>
> We apologize for absence of limitation discussions. While Domino ensures valid k-bFDR control under arbitrary dependence, this generality may come at a cost of power. Specifically, under independence or certain dependence structures, Domino does not fully exploit such information, reducing the ability to detect true alternatives. We will add this discussion to the revision.

---

> > ### Author Rebuttal · Reviewer_vcK7 · 2026-04-03
> >
> > I appreciate the authors’ efforts in preparing the rebuttal and clarifying several points. My overall assessment remains unchanged.

---

> > > ### Author Response · Authors · 2026-04-03
> > >
> > > Thank you for reviewing our paper. We would like to kindly ask whether there are any questions that we have not yet addressed. If so, we are happy to further discuss them with you.

---

### Decision · Program_Chairs · 2026-04-30

**Decision:**

Accept (regular)

**Comment:**

This paper studies the problem that standard multiple testing procedures often fail to guarantee the reliability of marginal discoveries at the boundary of the rejection set. Building on recent work on the boundary false discovery rate (bFDR), the authors propose k-bFDR, which bounds the joint error probability of the k least significant discoveries. Crucially, their method holds under arbitrary dependence.

All reviewers agree that this paper is timely, addresses an important problem, and is theoretically sound.

However, Reviewer pznL raised serious and valid concerns that must be addressed in the camera-ready version. Specifically, the reviewer rightfully observed that the initial experiments only reported how the True Discovery Rate (TDR) (and not the power!) changes as $\alpha$ varies. Additional experiments presented in the rebuttal showed that, while the TDR is very high, the power is often much lower. Hence the current experiments in the paper paint a much rosier. Furthermore, the reviewer noted that the signal strength in the original experiments was exceptionally large. The authors ran additional experiments under more moderate signal regimes to address this. The authors must ensure that all of these changes—specifically, the inclusion of the power metrics alongside TDR and the results from the moderate-signal-strength experiments—are explicitly incorporated into the camera-ready version.